# Reaction moments matter when designing lower-extremity robots for tripping recovery

**Saher Jabeen**[1], **Patricia M. Baines**[1]*, **Jaap Harlaar**[1,3], **Heike Vallery**[1,2], **Andrew Berry**[1,2]

**1** Department of Biomechanical Engineering, Delft University of Technology, Delft, The Netherlands,
**2** Department of Rehabilitation Medicine, Erasmus University Medical Center, Rotterdam, The Netherlands,
**3** Department of Orthopedics, Erasmus University Medical Center, Rotterdam, The Netherlands

☯ These authors contributed equally to this work.
* p.m.baines@tudelft.nl

**Data Availability Statement:** All simulation model and parameter files are available from the 4TUD data repository (https://doi.org/10.4121/21197149).

## Abstract

Balance recovery after tripping often requires an active adaptation of foot placement. Thus far, few attempts have been made to actively assist forward foot placement for balance recovery employing wearable devices. This study aims to explore the possibilities of active forward foot placement through two paradigms of actuation: assistive moments exerted with the reaction moments either internal or external to the human body, namely 'joint' moments and 'free' moments, respectively. Both paradigms can be applied to manipulate the motion of segments of the body (e.g., the shank or thigh), but joint actuators also exert opposing reaction moments on neighbouring body segments, altering posture and potentially inhibiting tripping recovery. We therefore hypothesised that a free moment paradigm is more effective in assisting balance recovery following tripping. The simulation software SCONE was used to simulate gait and tripping over various ground-fixed obstacles during the early swing phase. To aid forward foot placement, joint moments and free moments were applied either on the thigh to augment hip flexion or on the shank to augment knee extension. Two realizations of joint moments on the hip were simulated, with the reaction moment applied to either the pelvis or the contralateral thigh. The simulation results show that assisting hip flexion with either actuation paradigm on the thigh can result in full recovery of gait with a margin of stability and leg kinematics closely matching the unperturbed case. However, when assisting knee extension with moments on the shank, free moment effectively assist balance but joint moments with the reaction moment on the thigh do not. For joint moments assisting hip flexion, placement of the reaction moment on the contralateral thigh was more effective in achieving the desired limb dynamics than placing the reaction on the pelvis. Poor choice of placement of reaction moments may therefore have detrimental consequences for balance recovery, and removing them entirely (i.e., free moment) could be a more effective and reliable alternative. These results challenge conventional assumptions and may inform the design and development of a new generation of minimalistic wearable devices to promote balance during gait.

**Funding:** This research work (HV,SJ) was supported by the Netherlands Organisation for Scientific Research (NWO) Innovational Research Incentives Scheme Vidi grant 14865. The funders had no role in study design, data collection and analysis, decision to publish, or preparation of the manuscript.

**Competing interests:** The authors have declared that no competing interests exist.

## Introduction

The proportional and timely adaptation of foot placement is often a major contributor to balance recovery following a disturbance. Large perturbations at the support surface, such as tripping or stumbling, can require active foot placement or control over the dynamics of the swing leg to stabilise the trajectory of the center of mass [1]. During stepping, control of the movement of the swing leg also contributes to the regulation of angular impulse required for effective balance recovery [2]. To successfully avert a forward fall, the swing foot must be quickly placed anterior to the body and, in the process of doing so, must not make unintended contact with the ground. Muscle weakness and long reaction time can impair the ability to perform appropriate movements to recover balance following a perturbation, such as tripping over an obstacle [3]. Moreover, it has been shown that, compared to younger adults, older adults exhibit worse response inhibition for a task requiring step adjustment [4], which has been linked to an age-related decline in inhibitory abilities for choice of action [5]. Amongst community-dwelling individuals, more than 53% of falls occur due to tripping while walking [6, 7], and accidents on level ground account for half of all severe fall-related injuries [8]. People affected by stroke or other neurological disorders altering their muscle strength and/or response time also suffer from a particularly high risk of falls [9]. Therefore, older adults and persons with neurological impairment may benefit from an intervention that can improve foot placement by addressing or compensating for muscle weakness and slow response time.

Different interventions show promise for improving swing leg motion and foot placement. Gait training programs are effective at improving characteristics such as toe ground-clearance during the swing phase and can substantially reduce the chances of tripping or stumbling [10–12]. Amongst community-dwelling individuals, home-based therapy has the potential to incorporate such training into daily life with broad effect—for example lowering the fear of falling amongst frail older adults [13], improving functional gait scores amongst individuals with chronic stroke [14, 15], and reducing the rate of falls and near-falls amongst individuals with Parkinson's disease [16]. However, in reality little attention has been devoted to promoting motions necessary for effective balance recovery and ensuring safety during training in a community setting. Robotic orthoses could potentially provide emergency balance assistance for fall-prone ambulatory users during overground gait training or daily life. However, there is great diversity in how such a balance-modifying device might be realised [17]. A significant factor in this is how the actuator acts on the body to affect motion. Since forces and moments are always exerted in opposite action-reaction pairs, a robotic actuator exerting a force acting on a body segment must exert a reactive force elsewhere. From the point of view of the human body, the assistive action forces and moments are either internal, i.e., the corresponding reaction acts on another human body segment, or external, i.e., the reaction acts on another mechanical system.

Internal reaction systems, are highly favoured in contexts where mobility is a priority and often take the form of wearables such as rigid exoskeletons (EksoNR [Ekso Bionics Inc, Richmond, USA], Indego [Parker Hannifin Corp, Macedonia, USA], ReWalk [ReWalk Robotics Inc, Marlborough, USA]) or compliant tendon-driven exosuits (Harvard exosuit [18], Myosuit [19], ReStore [ReWalk Robotics Inc, Marlborough, USA]). The actuators are self-supporting or light enough to be carried, so that they are suitable for a wider range of overground environments than their non-wearable counterparts. While these designs have grown rapidly in popularity in recent years, it is poorly understood how the internal reactions might influence posture and balance. In particular, reaction forces on adjacent body segments have the potential to perturb posture unintentionally. Also, the fact that both 'action' and 'reaction' forces

act on the body means that forces cancel to a net-zero resultant in the absence of induced reactions at contact points with the environment. This complicates controlling whole-body properties important for balance, such as angular momentum [20, 21].

External reaction systems include treadmill-fixed gait trainers (Lokomat [22], LOPES [23]), rolling-frame gait trainers (KineAssist [24], Gable CORE [Gable Systems BV, Hengelo, NL]) and ceiling-fixed cable robots (Zero-G [25], FLOAT [26], RYSEN [27]). These can generate large forces or moments on the body to facilitate the movement of limbs or the body as a whole. However, their mass, size, and requirements for stable contact with the environment mean they are either constrained to a fixed workspace or are only mobile in large spaces with smooth surfaces. Emerging research explores the possibility of wearable systems with external reactions, such as reaction wheels [28], control moment gyroscopes [29, 30], and thrusters [31]. In such systems, moments are generated as a reaction to modifying the angular momentum of a spinning wheel or through the expulsion of a massive propellant.

While theoretical differences exist between these external- and internal-reaction actuation paradigms, it is unclear how they differ in facilitating balance recovery after perturbations like pushes, slips, or trips. In the case of a trip early in the swing phase, an effective way to recover balance in the sagittal plane is the 'elevation' strategy [32], a simultaneous lifting and anterior placement of the swing foot. A robotic aid might assist this motion by exerting a moment on either the thigh or the shank of the swing limb. Previous simulation results have suggested that both external- and internal-reaction moments acting on these body segments can, to different degrees, increase toe clearance and step length during unperturbed gait [29]. However, it is thus far unclear how such swing-leg assistance could be meaningfully utilised to recover balance following a perturbation, or how internal reaction moments might support or inhibit the recovery process.

The general objectives of this study are to (i) determine how a wearable robot might contribute to balance recovery following a perturbation, including the influence of its location on the lower extremities, and (ii) to assess the influences of internal reaction moments during actuation. To address these, we simulate and compare both external- and internal-reaction actuators and two different locations on the lower extremities, similar to our previous study [29]. A hypothetical wearable external-reaction actuator, hereafter referred to as a free moment (FM) actuator, is represented by an action moment applied to either the thigh or shank (Fig 1c). A comparable internal-reaction actuator spanning a single biological joint is represented by a similar action moment coupled with a reaction moment on the proximal neighboring body segment: the trunk or stance-leg thigh for swing-leg thigh assistance, or the thigh for shank assistance. We refer to this as a joint moment (JM) actuator, the conventional type for wearable exoskeletons.

Based on the theoretical differences of the actuator types and previous simulations of unperturbed gait [29], we hypothesise that

(H1) Both JM and FM actuators can beneficially assist balance recovery, but FM actuators are expected to do so more effectively and for a wider range of perturbations.

(H2) Internal reaction moments will perturb the kinematics of adjacent body segments. Depending on where and how these are applied, this will be associated with inhibited balance recovery. Specifically, we hypothesise (1) monoarticular joint moments assisting knee extension will simultaneously extend the hip and increase the likelihood of foot scuffing, and (2) joint moments assisting hip flexion will increase trunk forward lean and/or contralateral hip extension, which may counteract restabilization of the trunk necessary for successful balance recovery [33].

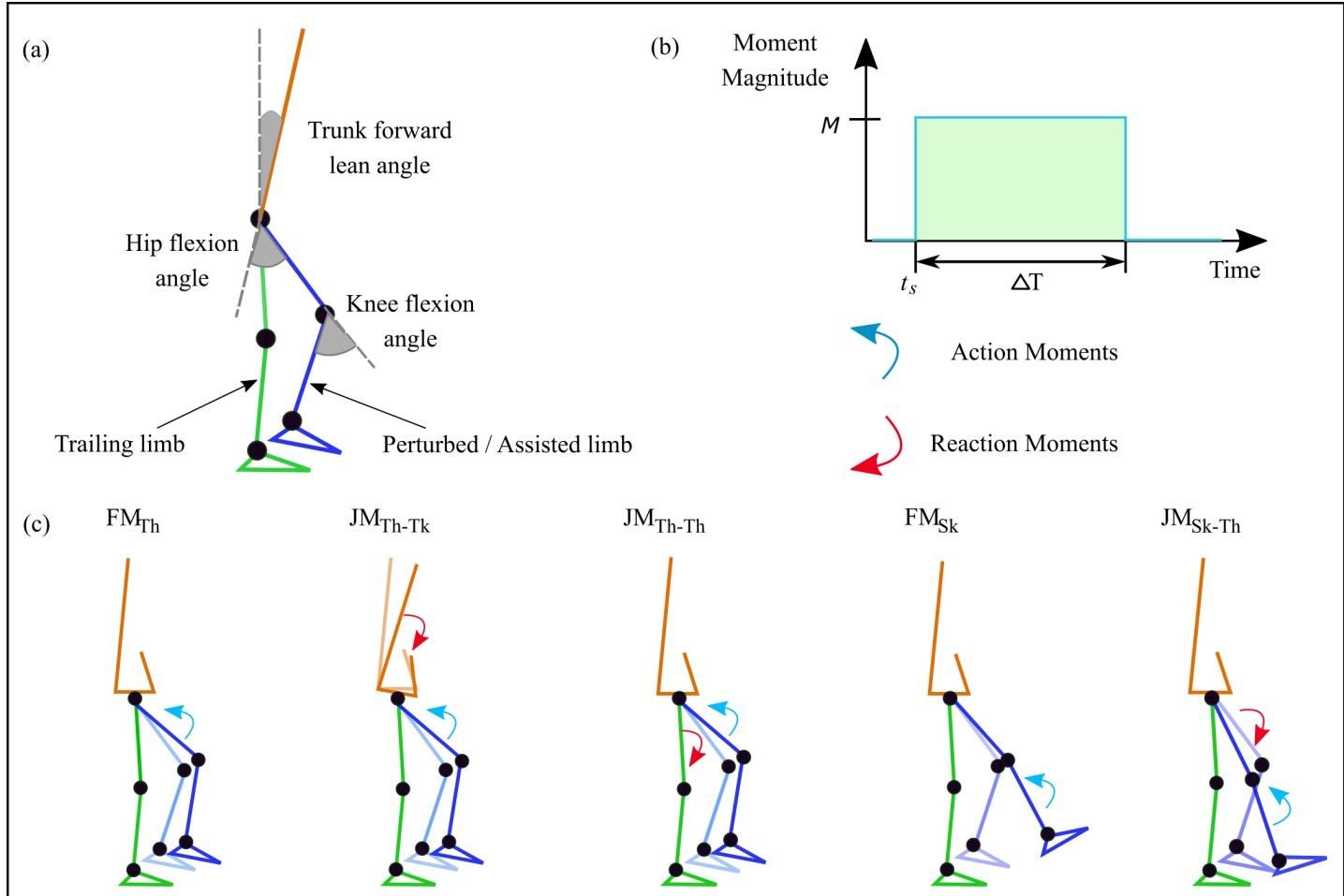

**Fig 1. Actuator and joint angle definitions.** (a) Definitions of joint angles of the sagittal-plane model. (b) The action moment is parametrised by its magnitude $M$, start time $t_s$, and duration $\Delta T$. The reaction moments equal in magnitude and opposite in direction to the action moments are applied to an adjacent limb segment. (c) Effects of the 5 different actuator types: free moment with action on thigh ($FM_{Th}$), joint moment with action on thigh and reaction on trunk ($JM_{Th-Tk}$), joint moment with action on thigh and reaction on contralateral thigh ($JM_{Th-Th}$), free moment with action on shank ($FM_{Sk}$), joint moment with action on shank and reaction on thigh ($JM_{Sk-Th}$).

(H3) Irrespective of the presence of a reaction moment, assisting movement of the thigh will result in more effective balance recovery than assisting movement of the shank. Previous simulations suggested that step length and toe clearance can be simultaneously increased through assisting the thigh [29], but the consequence for balance recovery has not yet been investigated.

Here, we quantify 'effectiveness' of balance recovery by the number of post-perturbation steps made prior to falling and the margin of stability (MoS) at the first recovery step. Joint angle trajectories additionally aid interpretation of the quality of recovery steps.

## Methods

### Simulation tool

To test our hypotheses we used the biomechanical simulation tool SCONE (version 0.21.0) [34]. The 2D (sagittal plane) lower-extremity musculoskeletal model contains 14 Hill-type musculotendon units configured according to Delp et al. [35] and using muscle dynamics

according to Millard et al. [36]. Each foot contains two contact spheres modeled with friction and restitution force. The reflex-based walking controller by Geyer and Herr [37] is used to generate walking patterns. The controller used to generate walking patterns has been widely validated in different studies to analyse aspects of impaired gait, such as the effects of aging on balance control [38, 39] and muscle activity [40]. The model has also been validated in response to mechanical disturbances, such as tripping by constraint of the swing leg and slipping of the stance leg, yielding response trends similar to experimental data for the majority of muscles [41]. The SCONE implementation of this model has also been validated against normative data [42]. More details of the model can be found in our previous work [29], and the model and controller parameter files available online [43].

To simplify the analysis, we assumed both hypothetical wearable actuator types have the same mass. To represent the actuator, we added 1.5 kg mass to both thighs for hip joint assistance or to both shanks for knee joint assistance. In both cases, the simulator's controller gains and initial conditions were re-optimised with the added mass for unperturbed gait with an average walking speed of 1 m/s.

Although the generated gait patterns were robustly stable to slight deviations, the walking controller was not optimised to recover the balance from large perturbations, and the model would fall if no intervention was made. This represents a scenario in which a human would need support to avoid falling.

### Perturbations

A spherical object of varying diameter (shape constrained by the software at the time of writing) fixed to a level ground surface was used to obstruct the swing leg and produce tripping events (Fig 2). The model walked approximately 22 steps (15 s of simulation time) both before and after obstacle contact to ensure that cyclical gait could be achieved if no fall occurred. The simulation model's gait initiation parameters were optimised for added mass to the legs but the gait controller was not optimised for the tested perturbations in order to emulate a naïve tripping response.

We analysed responses to five tripping perturbations with different obstacle sizes and times of contact within the gait cycle (Table 1). These perturbations (P1-P5) were heuristically chosen to produce a range of increasing perturbation severities—determined by the obstacle size and placement and foot-obstacle contact duration—to explore the feasibility limits of the different types of robotic assistance. To explore the efficacy of different interventions for preventing falls after tripping, the perturbations were designed such that the model could not sustain gait in the absence of external assistance.

In practice, tripping recovery strategies depend on the timing of the trip in the swing phase. We limit our scope to consider only trips occurring in the early swing phase (the first 20% of the gait cycle), which requires raising and anterior placement of the swing foot [32, 44]. All obstacles were placed accordingly. Obstacle height (diameter) was set between 5.5 cm to 9.5 cm and the duration of contact lasted between 40 ms to 150 ms. Perturbation P2 had the same height and placement as P1 but a higher dynamic friction coefficient to produce a longer duration of contact between the foot and obstacle. More details of each perturbation design and contact model can be found in the online repository [43].

### Assistance

We tested 5 different types of assistance in which either a flexion moment on the thigh or an extension moment on the shank was applied to assist forward placement of the foot of the obstructed leg. This assistance was applied unilaterally to the perturbed limb during the swing

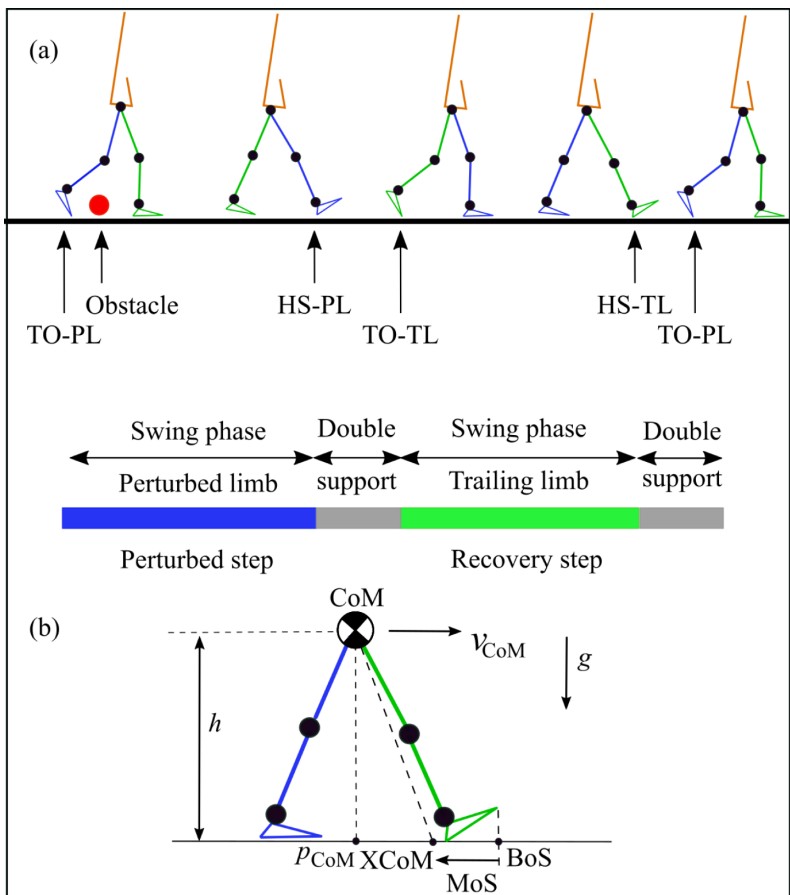

**Fig 2. Gait cycle and margin of stability definitions.** (a) TO-PL and HS-PL are toe-off and heel-strike events of the perturbed limb, and TO-TL and HS-TL are toe-off and heel-strike events of the trailing limb. The gait cycle is defined between two consecutive toe-offs of a perturbed limb (before and after hitting the obstacle). The perturbed step is the step in which the swing leg hits the obstacle. The recovery step is the step taken by the trailing limb after the perturbed step. (b) The margin of stability (MoS) is defined as the extrapolated center of mass position (XCoM) with respect to the position of the toe of the leading foot at heel-strike, the anterior extremum of the base of support (BoS). The anterior-posterior position of the CoM ($p_{\mathrm{CoM}}$) and velocity of the CoM ($v_{\mathrm{CoM}}$), height of the CoM above the ground ($h$), and gravitational acceleration ($g$) are used to calculate the XCoM.

**Table 1. Perturbation characteristics.**

| Pert | $H$ (cm) | $SP$ (%) | $\mu$ (-) | $D$ (s) |
|------|----------|----------|-----------|---------|
| P1 | 5.5 | 15 | 0.8 | 0.04 |
| P2 | 5.5 | 15 | 7 | 0.05 |
| P3 | 6.5 | 19 | 0.8 | 0.05 |
| P4 | 7.5 | 13 | 7 | 0.09 |
| P5 | 9.5 | 13 | 4 | 0.15 |

Different perturbation configurations (Pert) comprised a static spherical object of varied height ($H$), placement with respect to the swing leg (initial contact reported in swing phase, $SP$), and dynamic friction coefficient against the foot ($\mu$). These parameters were manually tuned to change the resulting duration of obstacle contact ($D$), which is approximately indicative of the magnitude of the balance perturbation.

phase of the obstructed step but after obstacle contact. For analysis, actuators are grouped in two: moment acting on the thigh (assisting the hip joint) and moment acting on the shank (assisting the knee joint). An illustration of the various types of assistance used in this paper is shown in Fig 1c.

For FMs, hip flexion is achieved with a moment acting on the thigh ($FM_{Th}$). In the case of a JM, the same moment acts on the thigh, but a reaction moment must be placed on another segment of the body. Depending on the design of an exoskeleton, the two conventional choices would be to either place the reaction moment on the pelvis/trunk ($JM_{Th\text{-}Tk}$), which is analogous to human monoarticular hip musculature, or on the thigh of the other leg (spanning both hip joints), which in our case is the trailing limb in stance phase during perturbation ($JM_{Th\text{-}Th}$) [45]. In this study, both cases of JM and FM for hip flexion are compared. For assisting knee extension, the assistive moment acts on the shank for both FM and JM assistance types. The reaction moment for JM is placed on the thigh, as it would be the most common choice for a knee exoskeleton and is analogous to the role of the monoarticular knee musculature.

For comparing actuator types and placement, all moments were parametrised as an ideal square wave of magnitude $M$, start time within the perturbed swing phase $t_s$, and sustained duration $\Delta T$ (Fig 1b). A range of values of these variables was simulated to determine the best parameters for each use-case. Considering the results obtained from a previous simulation study [29], the duration of the assistance was set to 200 ms, moment magnitude within the range [0, 20] N m with intervals of 2 N m, and start time in the range of [0, 210] ms with 30 ms intervals. Because the assistance is designed to reactively compensate for a balance disturbance, the start times are relative and to the moment that the foot clears the obstacle.

To compare the best-case realization of each actuator type and placement, an exhaustive grid-search was performed to determine the control parameters (assistance magnitude $M$ and start time $t_s$) that maximised the number of steps completed following the tripping event. The perturbation was applied halfway through a walking simulation of approximately 30 s, in which the latter 15 s for the model to get back in cyclic motion. If the model continued to walk for at least ten steps after obstacle contact, it was considered to have fully recovered—preliminary simulations showed that falls always occurred within five steps of a perturbation, similar to the number of steps required by healthy young adults to normalise gait following tripping [46]. In the event of multiple candidate parameter sets for a full recovery, the set that first minimised the assistance magnitude and then minimised assistance start time was selected (Fig 3). In the case of partial or failed recovery (the model walked only a few steps and then fell), the assistance parameter set was selected with maximum anterior margin of stability (MoS) [1] of the recovery step. Fig 2 shows the gait cycle and MoS definitions used for the analysis and the gait events in terms of perturbed and recovery limb.

## Performance measures

The primary criterion for assessing the efficacy of each intervention was whether or not the intervention prevented a fall following a tripping event. Although this is not a comprehensive metric of balance proficiency, it is a practical and unambiguous measure of *functional* performance with a high face validity for fall avoidance. Simulations in which a fall was successfully avoided were designated 'full recovery' (FR).

When a fall is not successfully avoided, the characteristics of the response are still relevant for analysis. Even if external assistance does not prevent a fall in simulation, it can mitigate the effects of the perturbation and prolong gait so that additional, real-world action could potentially be taken to prevent injury or sustain gait—for example, affording the wearer more time to react and execute their own postural response, complementary to the external assistance.

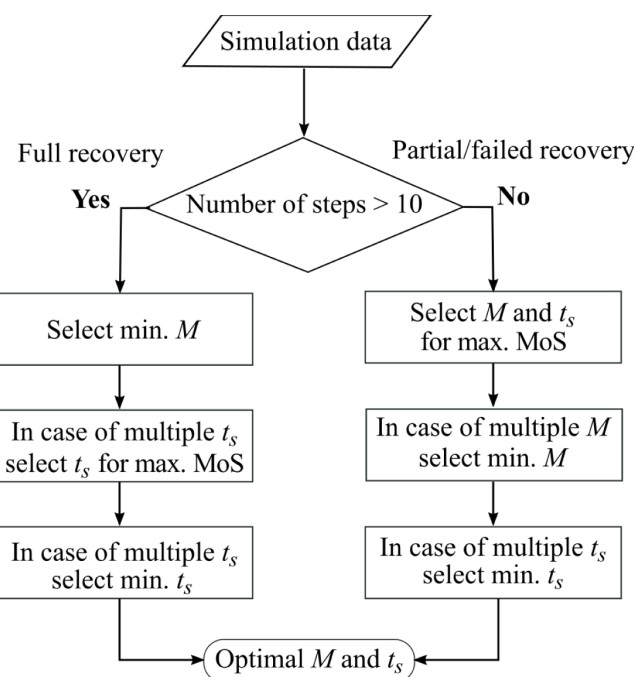

**Fig 3. Workflow for selection of best-case assistance parameters.** Selection procedure of assistance parameters magnitude $M$ and start time $t_s$ based on the outcome measures number of steps ($n$) and margin of stability (MoS) and the values of the assistance parameters.

Therefore, the number of steps between the tripping event and the falling event ($n$) was recorded as a secondary outcome. Steps were counted with respect to the first heel-strike after perturbation until either a body segment other than the foot contacted the ground or the heel of the swing foot was placed posteriorly to the toe of the stance foot.

To elucidate subtle differences between similar tripping responses, the maximum anterior margin of stability (MoS) of the first recovery step of the non-perturbed limb was also assessed. The MoS quantifies the (continuous) degree to which the CoP can be moved to restore static stability and is interpreted as a predictor of the robustness of balance [1]; negative values indicate that gait termination is not possible without an additional step. It is computed here as the position of the 'extrapolated center of mass' (XCoM, the position of the CoM adjusted for its velocity) with respect to the position of the toe of the leading foot at heel-strike (BoS) [47] (Fig 2b):

$$\text{MoS} = \text{BoS} - \text{XCoM} , \qquad (1)$$

where the XCoM is computed from the anterior-posterior position of the CoM ($p_{\text{CoM}}$) and velocity of the CoM ($v_{\text{CoM}}$), height of the CoM above the ground ($h$), and gravitational acceleration ($g$) [1]:

$$\text{XCoM} = p_{\text{CoM}} + v_{\text{CoM}} \sqrt{\frac{h}{g}} . \qquad (2)$$

The first step of the non-perturbed limb was selected for evaluating the MoS because it was found to be more correlated with the subsequent recovery steps than the MoS of the obstructed step. For a similar CoM velocity, increasing step length can increase the anterior MoS but decrease the posterior MoS (not reported here); hence, anterior MoS values that are either

much smaller or larger than those during normal gait can be associated with a higher probability of falling forward or backward, respectively.

The kinematic responses in the first step following perturbation were also analysed. Kinematic features such as the trunk forward lean angle and joint kinematics are widely used to differentiate between fallers and non-fallers [7, 33, 48]. Kinematic responses are reported for a gait cycle starting at toe-off of the perturbed (TO-PL) limb before perturbation and ending at toe-off of the same limb after perturbation (Fig 2a). The trunk forward lean angle is measured from the vertical axis and is reported as a positive angle when the body leans forward (Fig 1a). The hip flexion angle and knee flexion angle are presented along with the lengths of the first steps of the perturbed (swing) limb ($SL_P$) and the trailing (non-perturbed) limb ($SL_T$).

## Results

### Tripping recovery

The performance measures for all tripping obstacles, actuator types, and actuator locations are provided in Table 2. Unperturbed model continued to walk for the entire duration of the simulation. Without assistance (unassisted, UA), the model took a maximum of two steps when perturbed (all five perturbations). $FM_{Th}$ and $JM_{Th-Th}$ assisted perturbed gait successfully for P1, P2, P3 and P4 with maximum number of steps. In contrast, $JM_{Th-Tk}$ assisted full recovery from P1, P2 and P3 and partially assisted P4 with five steps only. $FM_{Sk}$ assisted full recovery for P1, P2 and P3. Amongst all perturbations, $JM_{Sk-Th}$ showed the best performance against P1 with only four steps completed following the tripping event. None of the interventions improved recovery for P5.

In response to P1, P2 and P3, assisting swing leg dynamics resulted in a MoS comparable to unperturbed gait and regained the gait stability for all types of actuators with moments acting on the thigh ($FM_{Th}$, $JM_{Th-Tk}$, and $JM_{Th-Th}$). Moments acting on the shank successfully restored balance in the case of FM actuators ($FM_{Sk}$) but not JM actuators ($JM_{Sk-Th}$). In response to P4, $FM_{Th}$ and $JM_{Th-Th}$ showed similar behavior and regained a MoS comparable to unperturbed gait. On the other hand, $JM_{Th-Tk}$ regained MoS but managed only five steps after the perturbation. In this case, step length of the trailing limb is comparably smaller than other successful trials (S1 Table). Unlike $JM_{Sk-Th}$, $FM_{Sk}$ marginally improved MoS but did not assist in

**Table 2. Performance of best case response to perturbations P1–5.**

| | Perturbation | P1 | | P2 | | P3 | | P4 | | P5 | |
| | Performance measure | n (steps) | MoS (m) | n (steps) | MoS (m) | n (steps) | MoS (m) | n (steps) | MoS (m) | n (steps) | MoS (m) |
|---|---|---|---|---|---|---|---|---|---|---|---|
| Assistance | UP | FR | 0.17 | FR | 0.17 | FR | 0.17 | FR | 0.17 | FR | 0.17 |
| | UA | 2 | -0.24 | 2 | -0.34 | 2 | -0.45 | 2 | -0.62 | 2 | -0.69 |
| | $FM_{Th}$ | FR | 0.17 | FR | 0.17 | FR | 0.17 | FR | 0.18 | 2 | -0.47 |
| | $JM_{Th-Tk}$ | FR | 0.16 | FR | 0.16 | FR | 0.16 | 5 | 0.076 | 2 | -0.6 |
| | $JM_{Th-Th}$ | FR | 0.17 | FR | 0.17 | FR | 0.17 | FR | 0.17 | 2 | -0.55 |
| | $FM_{Sk}$ | FR | 0.14 | FR | 0.15 | FR | 0.14 | 3 | -0.58 | 2 | -0.6 |
| | $JM_{Sk-Th}$ | 4 | -0.18 | 2 | -0.27 | 2 | -0.35 | 3 | -0.61 | 2 | -0.62 |

Performance evaluation of unperturbed gait (UP, for reference), unassisted gait (UA), free moment with action on thigh ($FM_{Th}$), joint moment with action on thigh and reaction on trunk ($JM_{Th-Tk}$), joint moment with action on thigh and reaction on contralateral thigh ($JM_{Th-Th}$), free moment with action on shank ($FM_{Sk}$) and joint moment with action on shank and reaction on thigh ($JM_{Sk-Th}$) for perturbation types P1-P5. Performance measures include the number of steps completed following the tripping event (n, where FR represents full recovery) and the margin of stability (MoS). The number of steps n includes the perturbed step and was found to be never lower than two.

**Table 3. Assistance parameters of best case response to perturbations P1–5.**

| | Perturbation | P1 | | P2 | | P3 | | P4 | | P5 | |
|---|---|---|---|---|---|---|---|---|---|---|---|
| | Assistance parameter | $M$ (N m) | $t_s$ (s) | $M$ (N m) | $t_s$ (s) | $M$ (N m) | $t_s$ (s) | $M$ (N m) | $t_s$ (s) | $M$ (N m) | $t_s$ (s) |
| Assistance | $FM_{Th}$ | 8 | 0.08 | 10 | 0.08 | 14 | 0.06 | 20 | 0.15 | 20 | 0.18 |
| | $JM_{Th-Tk}$ | 10 | 0.08 | 14 | 0.08 | 18 | 0.06 | 20 | 0.21 | 20 | 0.21 |
| | $JM_{Th-Th}$ | 8 | 0.08 | 12 | 0.05 | 14 | 0.06 | 20 | 0.09 | 20 | 0.06 |
| | $FM_{Sk}$ | 12 | 0.05 | 14 | 0.05 | 18 | 0.06 | 12 | 0.24 | 18 | 0.24 |
| | $JM_{Sk-Th}$ | 20 | 0.26 | 20 | 0.23 | 14 | 0.15 | 4 | 0.29 | 16 | 0.27 |

Assistance types: Assistance types: free moment with action on thigh ($FM_{Th}$), joint moment with action on thigh and reaction on trunk ($JM_{Th-Tk}$), joint moment with action on thigh and reaction on contralateral thigh ($JM_{Th-Th}$), free moment with action on shank ($FM_{Sk}$) and joint moment with action on shank and reaction on thigh (JMSk-Th). Assistance parameters: magnitude $M$ and start time ($t_s$), where $t_s$ is reported relative to perturbation start time. The assistance duration is fixed at $\Delta T = 0.2$ s for all cases.

regaining balance. P5 was the highest perturbation and no actuation type succeeded in regaining balance.

The assistance magnitude and start time of assistance for the best case recovery for each actuation and perturbation is shown in Table 3. When comparing the assistance magnitude of the $FM_{Sk}$ to $FM_{Th}$ it is shown that a larger moment is needed to recover balance in the case that the moment is applied on the shank. When comparing the assistance magnitude of $JM_{Th-Tk}$ to $JM_{Th-Th}$, a larger moment is needed when placing the reaction moment on the trunk. Further details of the dependence of $n$ on the assistance parameters is provided in Fig 4. Similarly, in Fig 5 it is shown that too high or too low values of MoS represents instability. For instance, in the case of $FM_{Th}$ and $JM_{Th-Th}$, assisting P1 with a higher magnitude of assistance (but the same onset time) resulted in partial recovery with a higher anterior MoS.

## Kinematics

Figs 6 and 7 show the kinematic response of all assistance types in response to perturbations P1 and P4, respectively. These two perturbations represent small and medium size tripping perturbations, respectively, and highlight the different postural requirements and influences of the actuator types. Kinematic responses to perturbations P2, P3 and P5 are provided in the supplementary material (S1–S3 Figs).

$JM_{Th-Tk}$ resulted in increased forward trunk lean angle compared to $JM_{Th-Th}$ and $FM_{Th}$. For P4 this increase of trunk lean angle persists during the recovery step. $JM_{Th-Th}$ and $FM_{Th}$ return the general response of the trunk lean angle back to the unperturbed gait for P1 through P4. Where in these cases the value does increase with increased perturbation, seen as an of-set in the trunk lean angle response. Similarly, $FM_{Sk}$ returns the general response of the truck lean angle back to the unperturbed gait for P1 through P3, also with an of-set in the value. On the other hand, $JM_{Sk-Th}$ is comparable to unassisted perturbed gait as it did not substantially extend $n$.

For moments acting on the thigh, early assistance cases resulted in an increased hip flexion angle and later assistance provided a similar hip flexion angle but extended the flexion phase compared to the unperturbed gait. For moments acting on the shank, $FM_{Sk}$ resulted in a slight decrease in hip flexion compared to the perturbed unassisted case. Similarly, in the case where $JM_{Sk-Th}$ slightly assisted the gait, it also reduced the hip flexion angle.

Moments acting on the thigh bring the general knee extension angle response back to the trajectory of unperturbed gait. The magnitude of the thigh assistance has no impact on the

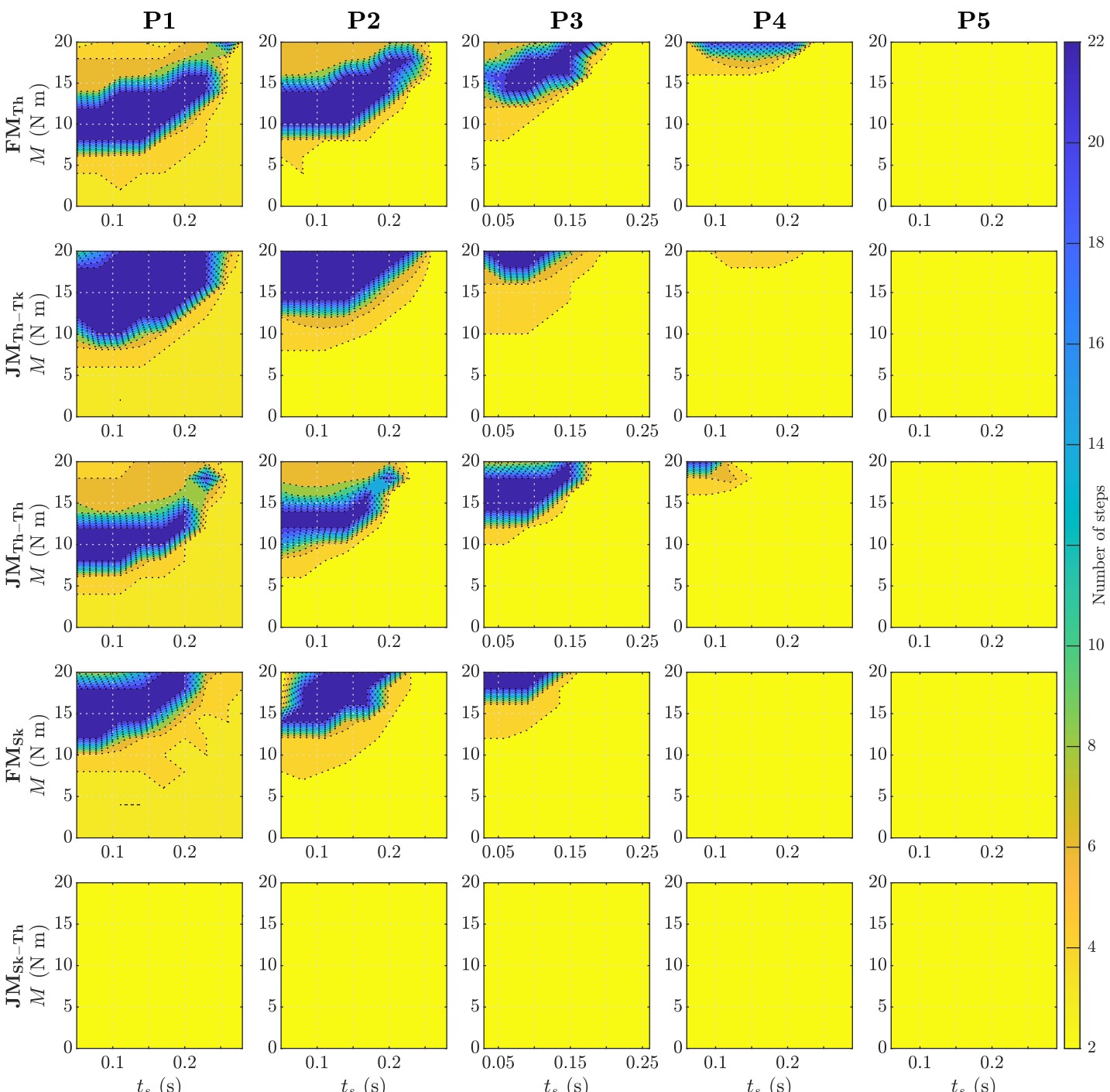

**Fig 4. Number of steps *n* as a function of assistance parameters (magnitude *M* and start time *$t_s$*).** The low end of the colour scale (pale yellow) represents unsuccessful recoveries in which model took only two steps before falling. The high end of the colour scale (dark blue) represents the maximum number of steps the model took in the unperturbed case (more than ten steps represents full recovery, FR). The recovery responses of a free moment with action on thigh ($FM_{Th}$), joint moment with action on thigh and reaction on trunk ($JM_{Th-Tk}$), joint moment with action on thigh and reaction on contralateral thigh ($JM_{Th-Th}$), free moment with action on shank ($FM_{Sk}$), and joint moment with action on shank and reaction on thigh ($JM_{Sk-Th}$) to perturbations P1–5 are shown.

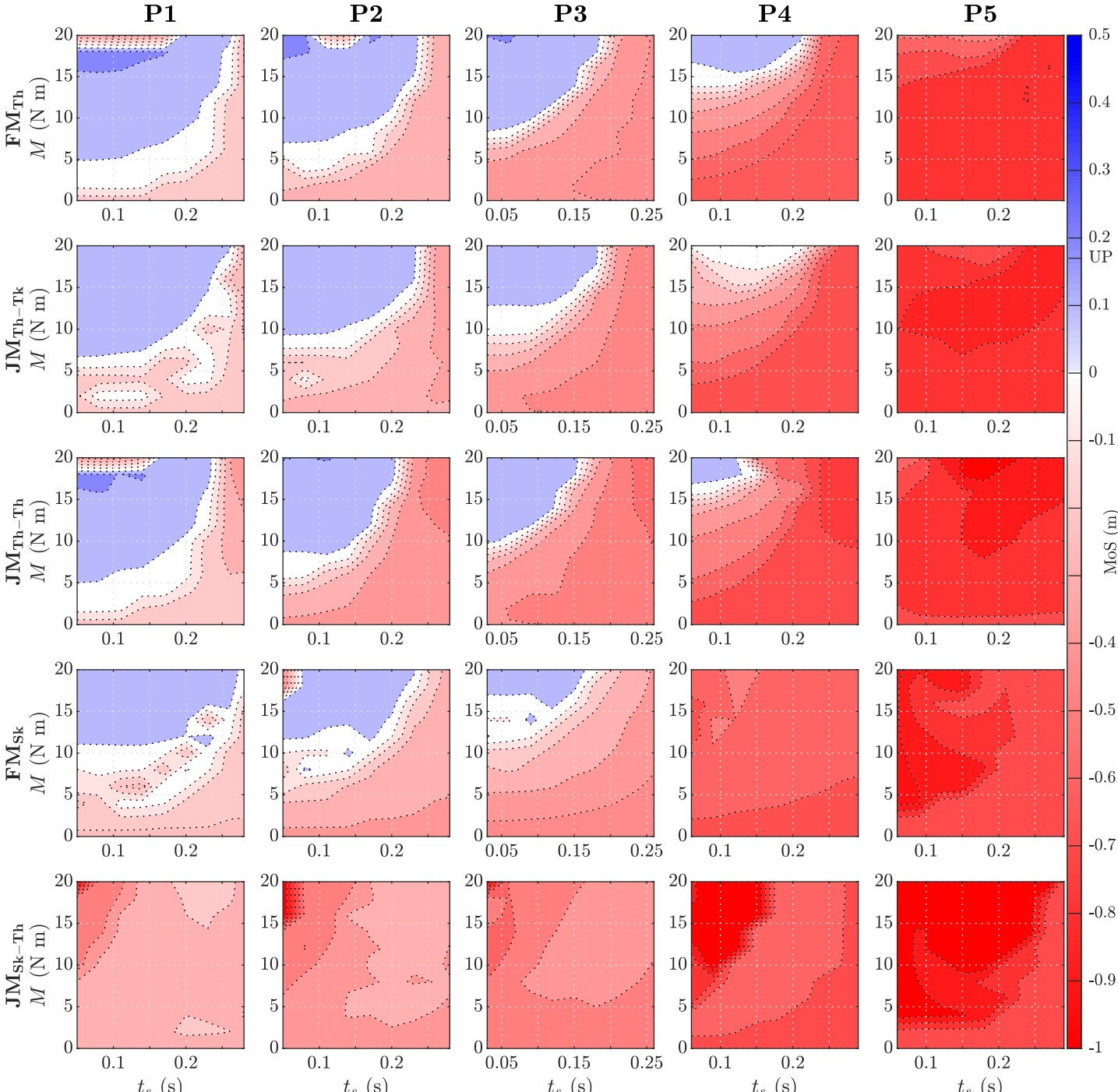

**Fig 5. Margin of stability (MoS) as a function of assistance parameters (magnitude *M* and start time *$t_s$*).** The minimum anterior MoS is taken at the heel-strike of the second step after perturbation. Negative values of the MoS are shown in red and positive values are shown in blue, with both colour intensities increasing with magnitude. The MoS during unperturbed (UP) gait is indicated in the colour scale on the right. The recovery responses of a free moment with action on thigh ($FM_{Th}$), joint moment with action on thigh and reaction on trunk ($JM_{Th\text{-}Tk}$), joint moment with action on thigh and reaction on contralateral thigh ($JM_{Th\text{-}Th}$), free moment with action on shank ($FM_{Sk}$), and joint moment with action on shank and reaction on thigh ($JM_{Sk\text{-}Th}$) to perturbations P1–5 are shown.

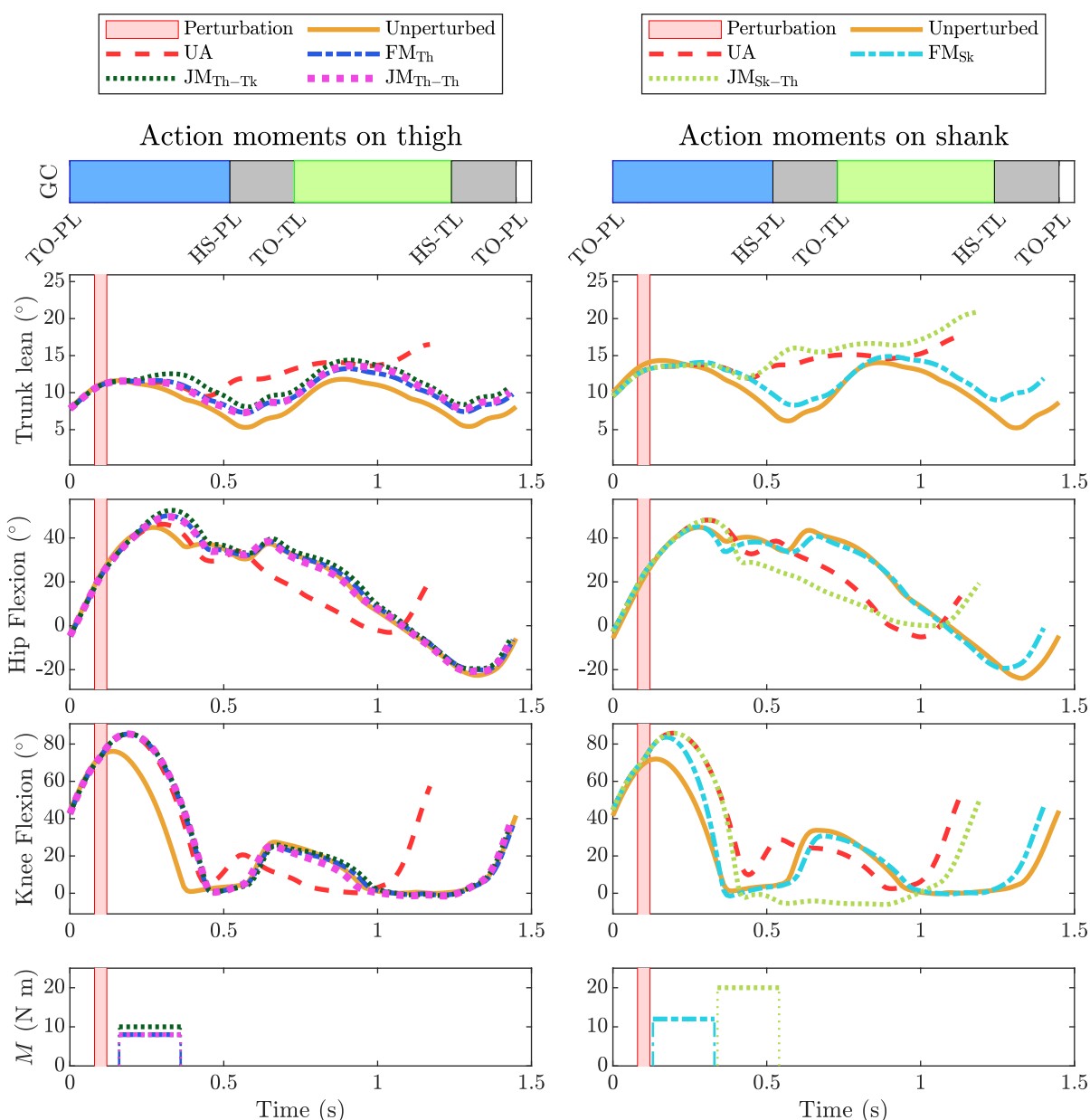

**Fig 6. Kinematic response to perturbation P1.** Left column: Assistance via hip flexion of the perturbed limb. Conditions denoted by lines are unperturbed (UP), unassisted (UA), free moment with action on thigh ($FM_{Th}$), joint moment with action on thigh and reaction on trunk ($JM_{Th\text{-}Tk}$), and joint moment with action on thigh and reaction on contralateral thigh ($JM_{Th\text{-}Th}$). Right column: Assistance via knee extension of the perturbed limb. Conditions denoted by lines are unperturbed (UP), unassisted (UA), free moment with action on shank ($FM_{Sk}$), joint moment with action on shank and reaction on thigh ($JM_{Sk\text{-}Th}$). Individual plots show the trunk lean angle, hip flexion angle, and knee flexion angle, and applied moment $M$ as a function of time. Bar graphs at top of each column show the gait events of the unperturbed gait cycle (GC), where TO-PL and HS-PL are toe-off and heel-strike events of the perturbed limb, and, TO-TL and HS-TL are toe-off and heel-strike events of the trailing limb.

knee extension angle, as it follows the same trend for all cases; as seen in response to P2, all case of thigh assistance have a different assistance magnitude but no difference can be seen in the knee extension angle (S1 Fig). A slight difference is visible in P4, where one may notice the delayed knee extension (Fig 7) which is the result of delayed start of assistance. Moments

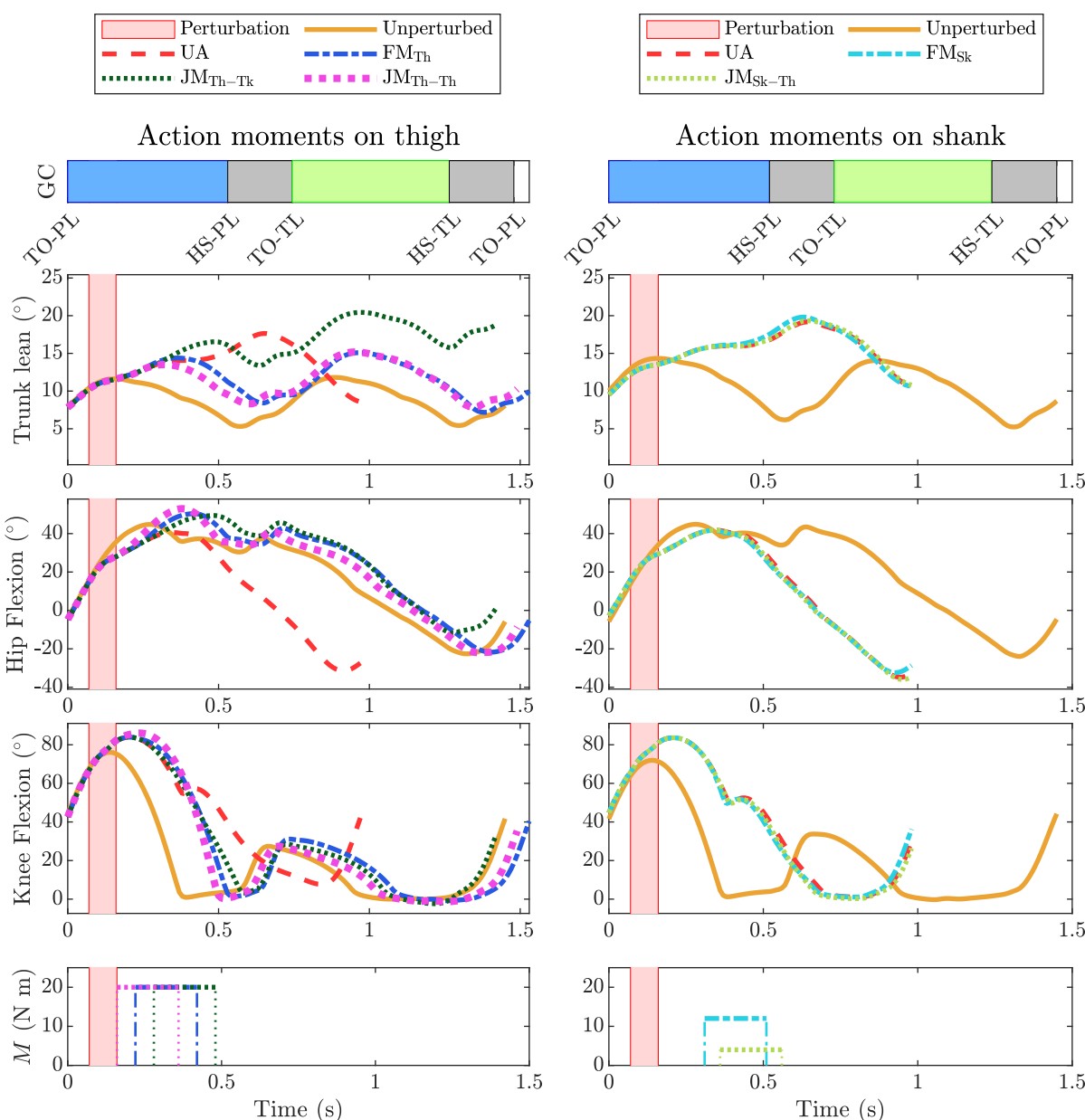

**Fig 7. Kinematic response to perturbation P4.** Left column: Assistance via hip flexion of the perturbed limb. Conditions denoted by lines are unperturbed (UP), unassisted (UA), free moment with action on thigh (FM_Th), joint moment with action on thigh and reaction on trunk (JM_Th-Tk), joint moment with action on thigh and reaction on contralateral thigh (JM_Th-Th). Right column: Assistance via knee extension of the perturbed limb. Conditions denoted by lines are unperturbed (UP), unassisted (UA), free moment with action on shank (FM_Sk), joint moment with action on shank and reaction on thigh (JM_Sk-Th). Individual plots show the trunk lean angle, hip flexion angle, and knee flexion angle, and applied moment $M$ as a function of time. Bar graphs at top of each column show the gait events of the unperturbed gait cycle (GC), where TO-PL and HS-PL are toe-off and heel-strike events of the perturbed limb, and, TO-TL and HS-TL are toe-off and heel-strike events of the trailing limb.

acting on the shank have a clear impact on the knee extension angle, as it increases the knee angle magnitude and the duration compared to unassisted perturbed gait. In the event of a successful recovery, FM_Sk returns the general response of the knee angle back towards the unperturbed condition.

## Discussion

Simulations were conducted to investigate the influence of internal and external reaction moments on posture and balance recovery following tripping, as well as the suitability of different actuator placements on the segments of the lower extremities. Differences in balance recovery performance were observed with regard to both actuator type and location. At least one realization of both actuator paradigms were capable of stabilising the model following perturbation scenarios P1-P4. The inability of any actuator to stabilise P5 may indicate that a larger moment magnitude may be required (stabilising solutions in P4 approach the 20 N m moment magnitude boundary, as shown in Fig 4, representing the number of steps completed following the tripping event ($n$) in response to assistance parameters) or a different assistance strategy must be adopted. We observed large differences in recovery responses between different placements of the reaction moments.

### Monoarticular reaction moments reduce the efficacy of assistance (H1)

**The greatest functional differences of reaction moments occur on the shank.** Although both actuator paradigms were capable of stabilising the model after tripping using the selected recovery strategy, this was not the case for both actuator locations. The presence of a reaction moment produced substantial functional differences while actuating the shank. While $FM_{Sk}$ was able to successfully assist recovery from perturbations P1-P3, $JM_{Sk-Th}$ was unable to prevent falling in any of the cases. In contrast, reaction moments appeared to only modestly affect actuating the thigh, wherein $FM_{Th}$, $JM_{Th-Tk}$, and $JM_{Th-Th}$ yielded similar $n$ and anterior MoS in perturbations P1-P3 and deviated only in P4. While exerting a reaction moment on the trunk appears to have a negative effect on balance recovery ($FM_{Th}$ was able to stabilise P4 but $JM_{Th-Tk}$ was not), exerting it instead on the contralateral thigh ($JM_{Th-Th}$) had a less obvious detriment that was discernible only by a lower MoS (Table 2 and Fig 5) and the smaller set of feasible parameters (Fig 4) in P4. However, the existence of feasible solutions for both actuator paradigms for hip flexion assistance suggests that the presence of reaction moments on the body is neither a fundamental requirement nor barrier to the assistance of balance recovery.

**Assistance parameters reveal differences in model sensitivity.** In instances where actuators had a similar functional result (i.e., recovered gait), they can still be discriminated by subtle differences in the sensitivity of the model to the assistance parameters. The location and size of the contour regions of $n$ in Fig 4 (and to a lesser extent Fig 5) indicate the sensitivity and robustness of successful stabilization to the assistance magnitude and onset time. With respect to the parameter ranges explored, there appears to be considerable tolerance to the assistance onset time—most early-mid values yielded similar results within and between perturbations and amongst all actuator types and locations (with the exceptions of P5 and $JM_{Sk-Th}$, which did not yield feasible results). The feasible moment magnitude, on the other hand, is confined to a smaller range with both lower and upper bounds (associated with insufficient and excessive anterior MoS, respectively, Fig 5), and shifts appreciably with perturbation magnitude. Compared to a free moment applied to the thigh ($FM_{Th}$), the minimum moment required to stabilise the model was 25–40% larger when a reaction moment was present on the trunk ($JM_{Th-Tk}$, Table 3) but similar when the reaction moment was instead exerted on the contralateral thigh ($JM_{Th-Th}$).

**Location of reaction moments can determine whether balance recovery is feasible.** While both actuator paradigms can be useful in the appropriate context, these simulations suggest that, for the application and control scheme presented here, internal reaction moments can either catastrophically inhibit performance (shank) or decrease sensitivity to the assisting moment (thigh). However, the simulations of the thigh assistance also show that it is possible

to mitigate this interference through judicial placement of the reaction moment (i.e., moving the reaction from the trunk to the contralateral thigh). Hence, care must be taken when designing an internal-reaction actuator. Prior to considering other (practical) factors pertaining to the design objective, external-reaction actuators may offer a conceptually simpler alternative for controlling body segment motion for high-level tasks like balance control.

### Internal reaction moments alter body kinematics (H2)

**Knee extension reaction moment induces early foot contact.** Both actuators with an action moment on the shank ($FM_{Sk}$ and $JM_{Sk-Th}$) extend the knee joint (Fig 6) with the intention of bringing the foot forward with respect to the knee and increasing anterior foot placement; however when a reaction moment is simultaneously applied to the thigh ($JM_{Sk-Th}$), hip flexion is inhibited during the mid-swing phase and/or hip extension occurs prematurely during the late-swing, depending on when the assistance is initiated. This decreased flexion or increased extension lowers the knee with respect to the pelvis and, consequently, lowers also the foot and causes early ground contact. During the swing phase, the pelvis forward velocity is not substantially affected by the assistance, so early swing termination decreases step length (S1 Table) and creates a smaller anterior moment arm between the CoP and CoM, lessening the centroidal retarding moment needed to counteract the effect of the perturbation. This is evidenced by a decreased anterior MoS and number of successful steps when compared to $FM_{Sk}$ in several of the perturbation conditions (Table 2, Figs 4 and 5) and the finding that $JM_{Sk-Th}$ is often most effective when the assistance is too low or late to influence foot placement (Fig 5).

Incidentally, if cadence remains similar, a decreased swing time and step length due to knee joint assistance may also explain the empirical finding that unimpaired users of powered knee orthoses do not walk faster than when not wearing a device, contrary to the expectation that this would augment physical performance (although it is unclear to what extent this may also be due to the added inertia or impedance of the device) [49].

From the simulated scenarios, a conclusion cannot be drawn on whether $JM_{Sk-Th}$ is an invalid mode of assistance for tripping recovery, or whether success was limited by the control scheme or moment constraints. An alternative control scheme might first flex the knee to increase toe clearance before extending the knee to increase step length.

**Hip flexion reaction moment destabilises trunk posture.** All three actuators with an action moment on the thigh ($FM_{Th}$, $JM_{Th-Tk}$ and $JM_{Th-Th}$) flex the hip; however, when the reaction moment is applied on the trunk, as with a traditional hip exoskeleton ($JM_{Th-Tk}$), there is also an increase seen in the forward trunk lean angle. This effect on the trunk lean angle increases as the magnitude of the assistance increases for larger perturbations. This is an important trend, since kinematics elements such as the trunk lean angle can be associated with imbalance after perturbation [33]. The fact that balance was only partially recovered after P4 by $JM_{Th-Tk}$, whereas $FM_{Th}$ resulted in a full recovery, also points at the destabilising effect of the trunk reaction moment. Even if the effect on forward trunk lean does not directly cause instability, it is important to consider how the user would perceive such an effect. Among other factors, control of the trunk angular velocity is one of the critical factors in balance recovery [33], especially in elderly people [48]. Adding moments to trunk in the direction of fall is likely to hinder in a person's own active control of their trunk angular velocity.

**Hip flexion reaction moment on contralateral thigh is a good alternative.** The actuators with action moment on the thigh and external reaction moment ($FM_{Th}$) and internal reaction placed on the contralateral thigh ($JM_{Th-Th}$) show very similar performance and kinematics. The reaction moment on the contralateral thigh could promote trailing limb hip extension,

which could also promote the forward walking progression. This is seen here as a small effect seen in the trailing limb kinematics (S4 Fig), namely a small extension of the hip and knee, which likely also slightly increases push-off. We therefore propose that the reaction moment placement on the contralateral thigh is a superior alternative to placing it on the trunk.

### Actuating the thigh promotes both hip and knee dynamics necessary for balance recovery (H3)

**Assisting thigh or shank affects gait kinematics.**   When unassisted, the perturbation causes the swing foot to contact the ground prematurely. Contacting the ground early does not allow the knee time to fully extend, which substantially shortens the step length. Actuating the thigh flexes the hip and lifts the knee, increasing foot clearance and extending the swing time with respect to the unperturbed case. This lengthened swing phase gives the knee sufficient time to fully extend, increasing anterior foot placement. On the other hand, assisting knee extension directly does not increase the swing time with respect to the unperturbed case but instead increases knee extension velocity, bringing the knee to full extension quicker (Fig 6). In the cases where both strategies successfully assist recovery, the step lengths are similar— substantially larger than the unassisted case but smaller than the unperturbed case.

In the cases where the FM action on the shank successfully assists recovery, hip extension is also indirectly affected, even though there are no moments exerted on the thigh as in $JM_{Sk-Th}$. The hip briefly extends as the knee reaches full extension, which may be the result of the reflex model reacting to increased swing leg velocity by increasing hip extension near the end of swing phase. Similar behavior is seen in the unperturbed case, as the hip extends to retract the swing leg before making foot contact. Moreover, such increase in hip extension could also be facilitated indirectly by knee joint reaction forces. When the knee is extending, a reaction force at the knee joint would extend the hip if the knee is not stiff.

**Assisting hip flexion out-performs assisting knee extension.**   For both actuation paradigms, actuating the thigh was more likely to result in successful tripping recovery and require a smaller magnitude of applied moment to do so. In the case of internal-reaction actuators, actuating the shank ($JM_{Sk-Th}$) was not capable of stabilising the model for the given perturbation and assistance magnitudes, while both realizations of thigh actuators ($JM_{Th-Tk}$ and $JM_{Th-Th}$) performed relatively well. Similarly, external-reaction actuators at the thigh ($FM_{Th}$) required 20–30% less assistance than an identical actuator placed on the shank ($FM_{Sk}$) for perturbations P1-P3 (Table 3).

### Practical application

**Assisting subfunctions for tripping recovery.**   These simulations show that multiple strategies are feasible for assisting balance recovery following tripping—protectively increasing step length can be achieved by either increasing the rate of knee extension or prolonging the swing phase by increasing hip flexion. However, despite yielding similar step length, they produce different trajectories of the swing foot. Previous simulations indicate that augmenting knee extension before mid-swing decreases the ground clearance of the foot when it is at its highest velocity [29], which can increase the risk of tripping over obstacles or uneven terrain or result in foot scuffing over even smooth surfaces. While we have thus far assumed a single discrete tripping obstacle, it is conceivable that prematurely assisting knee extension could even impair tripping recovery if presented with further variations in surface height. Either care must be taken to ensure that knee-extending assistance is initiated only after mid-swing, or the assistance should bimodally first flex the knee to increase foot clearance before switching to extension. In either case, accurately estimating the gait phase and synchronising the assistance

may be critical. In contrast, the same previous simulations showed that assisting instead hip flexion in the early swing phase both lifts and flexes the knee and increases ground clearance while simultaneously increasing step length [29]. This may constitute the more robust and safer alternative.

While these simulations show that it is possible to influence step length from either the shank (knee) or thigh (hip) independently, it may also be possible to combine these strategies. Placing multiple smaller actuators on the legs could allow the knee and hip joints to be independently manipulated but, in the context of providing only emergency assistance, the obtrusiveness of the larger number of points of contact and greater overall mass may detract from any functional benefit. Alternatively, the hip and knee degrees of freedom might be coupled to mimic the functions of the biarticular leg muscles [50], which has been investigated for facilitating steady locomotion and sitting transfers (e.g., [19, 51]) but not yet for balance control.

**Actuator placement for tripping recovery.** Our simulations suggest that actuators on the hip or thigh would likely not need to exert moments as large as those placed on the knee or shank for a similar degree of functional support ($n$ or MoS), which could potentially entail smaller and lighter actuators. For a wearable aid, it is generally advisable to not only minimise the amount of mass added to the body, but also minimise its distance from the center of mass —placing mass distally on the lower extremities can undesirably increase both the metabolic cost of walking [52] and the rotational inertia of the leg about the hip joint, which may reduce the efficacy of assistance or require more powerful (and potentially heavier) actuators. These factors visibly influence the design of many existing minimally-obtrusive wearable robots, for example, recent tendon-driven JM systems in which the actuators and power supplies are placed near the body's center of mass [18, 19].

Placing an actuator distally on the leg could instead be preferable if it could be integrated with an existing device worn by the user, for example, incorporating an FM actuator into an ankle orthosis or a transtibial prosthesis, or either a JM or an FM actuator into a transfemoral prosthesis.

**Internal-reaction wearable actuators.** The vast majority of existing wearable robots use internal-reaction actuators. Monoarticular joint actuators, in which each biological degree of freedom is controlled by a separate dedicated actuator, have been conventionally preferred in the design of exoskeletons. Perhaps because these systems have traditionally been designed to primarily oppose gravity and facilitate movement rather than specifically regulate balance, relatively little attention has been paid to how reaction moments affect stability during dynamic activities. The simulations presented here suggest that such architectures can undesirably influence passive mechanics during gait subfunctions and be disadvantageous for balance control. Contrary to most state of the art, if it is assumed that the user is independently ambulatory and the robotic orthosis is *exclusively* for emergency balance correction, then there is also a potential usability drawback of such a design.

Recent trends in tendon-driven exosuits have drawn inspiration from biological multiarticular musculature to offer greater flexibility in addressing these concerns. Multiarticular muscle-tendon complexes span and actuate multiple joints simultaneously, simplifying neurological control and power transmission from powerful proximal muscle groups to distal joints [50]. Similarly, minimalistic wearable robots can employ a small number of favourably-placed artificial tendon actuators to support a core set of subfunctions necessary for gait without actuating or constraining all leg joints [50, 53, 54]. Significantly, this yields greater freedom in specifying the locations of the action and reaction forces on the body, and enables heavier components to be placed close to the CoM to minimise energetic cost and disturbance to natural gait patterns. The actuator JM$_{\text{Th-Th}}$ simulated here represents an uncommon parallel biarticular construct without a direct biological counterpart. However, a similar robotic hip orthosis

using instead two coordinated monoarticular actuators has previously shown great potential for aiding balance recovery following slipping [45].

**External-reaction wearable actuators.**   From a theoretical standpoint, our simulation results appear to favour the use of external-reaction actuators for this particular use-case. Although the hypothetical actuators have thus far been idealised as a mass and moment source(s), their physical realisation can take various forms with different practical consequences. We define here two subclasses of wearable external-reaction actuators: contact-manipulating actuators, such as active ankle orthoses [55, 56] and supernumerary limbs [57] that exert forces directly against the environment, and inertia-manipulating actuators, such as artificial tails [58], resonating pendula [59], reaction wheels [28], control-moment gyroscopes [30], and cold-gas thrusters [31] that induce moments or forces by manipulating the motion of a mass contained within (or, in the latter case, expelled from) the actuator itself.

Contact-manipulating actuators are capable of generating relatively large, sustained forces to support balance and gait. However, a constraint is that this is realised through a (temporarily) stable mechanical connection with the environment and is therefore effective only when the actuated limb is in stance and on surfaces that are preferably neither compliant nor slippery. The design and manner of achieving contact with the surroundings can also influence compatibility with different terrains (e.g., stairs) and the likelihood of tripping over components or inhibiting natural stepping patterns.

Inertia-manipulating actuators, in contrast, do not require ground contact and can assist body movement even if contact with the surroundings is limited or unreliable. Nor do these actuators necessarily need to be placed around the feet to influence balance (in fact, in most examples they are not even on the lower body [28, 30, 31]), thus preventing issues with tripping or terrain compatibility. There is, in principle, considerable freedom in where and how these actuators attach to segments of the body, since they do not need to span the biological joints to exert reaction moments. The relatively few points of contact with the body simplifies the donning and doffing of the device and reconfiguration of the interface for other users. However, the principle of operation also introduces several limitations. The presence of a passive mass for energy storage, and sometimes multiple motors (e.g., control moment gyroscopes), can entail that inertial-manipulating actuators are often larger and heavier than alternatives with a similar power output. While the detrimental effects of added mass on gait is similar for all wearable devices, actuators with rotational energy storage (e.g., reaction wheels and control moment gyroscopes) also have the potential to disturb gait through unintended gyroscopic moments induced through rotational movement of the actuated body segment—however, these induced moments can be partly or wholly cancelled internally via mechanical constraints [60] or control [61] or even be exploited to produce intentional changes in gait patterns [62]. Finally, although all actuator types are limited in their maximum output moment and power, inertia-manipulating actuators also have impulse constraints that limit the duration of moment application and prevent them from providing continuous support in static directions.

Given the relatively recent appearance of these technologies in wearable applications, there are few empirical analyses and it is yet unclear which actuation architecture is best (or even feasible) for practical implementation. For example, most empirical studies involving control moment gyroscopes have thus far sought to influence balance through control of the motion of the CoM [30, 63, 64], whereas control of the lower extremities has been investigated only in simulations [65]. However, parallel research has explored the uses of these actuators on the limbs for other purposes, such as tremor-suppression (GyroGlove by GyroGear; London, UK), upper-extremity prostheses [66, 67], and emulation of virtual environments [68]. Ongoing research continues to investigate the extent to which optimisation of mechanical designs can

mitigate technical limitations and exploit the proposed advantages [69]. While much development and validation of these emerging technologies remains, they may soon fulfill a complementary roll in the technological landscape and offer versatile solutions for facilitating limb movement and balance.

**Potential beneficiaries.** The target group is one of the essential factors to consider when designing an assistive device. We believe that elderly, people suffering from stroke and neuromuscular issues could benefit from forward foot placement assistance. Although in this study we only investigated the forward foot placement assistance, a FM device placed on either the thigh or shank could also assist in ab/adduction and endo/exorotation of the hip. For instance, it is commonly seen that people suffering from stroke circumduct the swing leg as a compensatory movement to avoid foot scuffing. Moreover, ad/abduction assistance can compensate for weaker muscles which otherwise results in scissored gait in patients with cerebral palsy.

## Study limitations and future work

The SCONE planar simulation model and underlying reflex model has been previously validated against human data for unperturbed walking and certain types of balance disturbances [42]. However, it is unclear the extent to which dissimilarities in the experimental protocols (e.g., the mode, magnitude, and timing of balance disturbance) and different choice of outcome measures would affect the model validity. For example, trunk kinematic responses are critical for distinguishing fallers from non-fallers during tripping responses [48], yet here the upper body was modelled as a single segment, in which the pelvis, vertebrae, head, and arms were rigidly fused together. Qualitative comparison of our simulation results (unassisted condition) with empirical studies show a similar trend between peak trunk lean angle and the magnitude of the tripping perturbation [33, 70] and similar characteristics of the knee and hip joint responses following tripping [33]. Nevertheless, a quantitative comparison is lacking and there is currently no data to validate the neuromechanical response to external moments applied at different locations on the body. It is thus far unclear whether moments produced by wearable actuators could, under any circumstances, initiate additional, unmodelled reflexes that might change the response characteristics. In addition, a JM hip actuator would conventionally attach to the thigh and pelvis, but, because the pelvis and trunk are fused in the simulation model, the simulation may falsely exaggerate trunk lean when assistive moments are applied.

In reality, trips and balance recovery can occur in many more ways than were explored in this study [32, 71] and, therefore, these results might not be generalizable to other trip types. In particular, we investigated tripping only in the early swing phase, which emphasises a response strategy in which the perturbed foot is elevated and placed anteriorly to the CoM to arrest forward momentum. In contrast, tripping in late swing phase can be best overcome by placing the perturbed foot down and taking another step with trailing limb, called the 'lowering' strategy [32]. In such a situation, assisting swing leg motion might have limited benefit and result in negligible kinematic changes due to the short swing duration. Although not investigated here, it is possible that assisting the same leg subfunctions may be useful to recover from other modes of disturbances. Forward and backward pushes to the pelvis might similarly require assisting hip extension during the late swing phase to halt motion of the swing leg [72], and slipping might be mitigated by assisting hip flexion and extension in the double-support phase [45].

In practice, determining an 'optimal' magnitude of assistance can be challenging due to the dependence on both context-specific factors (e.g., size of perturbation, walking speed, gait phase) and user-specific factors (e.g., body inertial properties, reaction time, strength,

coordination). While it may not be feasible to model each user and scenario perfectly, a sensitivity analysis of independent variables and identification of their expected ranges could inform future design work. For example, the control action might scale with respect to a minimal set of independent variables and still be robust to the majority of use-cases without having to measure all variables and model all dynamics.

Additional points of interest for further study include incorporating actuator dynamics and constraints in the simulator, investigating other actuator configurations (e.g., multiarticular actuation of the knee and hip), and exploring other control objectives (e.g., continuous assistance to reduce fatigue or compensate for the energetic cost of carrying the mass of the actuators).

## Conclusion

This simulation study analysed the effectiveness of free moments and joint moments to recover balance after tripping while walking. Both free moments (with external reaction moments) and joint moments (with internal reaction moments) can effectively assist balance recovery. This investigation leads us to conclude that reaction moment placement for joint moment devices impacts gait dynamics and needs to be carefully considered when designing an assistive device. Reaction moments placed on the trunk or pelvis for hip flexion assistance resulted in increased trunk forward lean angle that unfavourably affects balance recovery if the person is falling forward. Hence, one may consider putting the reaction moments for hip flexion assistance on the contralateral thigh. Assisting knee extension using joint moments with reaction moments placed on the adjacent thigh induces early foot contact and makes forward foot placement for balance recovery ineffective. We also concluded that free moments might provide more flexibility in application and wearability. Unlike joint moments, free moments are not only effective in balance assistance when placed on the thigh, but they can also effectively assist balance recovery when placed on shank. Overall, assisting hip flexion for forward foot placement outperforms assisting knee extension as it promotes both hip and knee dynamics required for effective balance recovery.

## Supporting information

**S1 Table. Step length of best case response to perturbations P1-5.** Step length response corresponding to unassisted gait (UA), free moment with action on thigh ($FM_{Th}$), joint moment with action on thigh and reaction on trunk ($JM_{Th-Tk}$), joint moment with action on thigh and reaction on contralateral thigh ($JM_{Th-Th}$), free moment with action on shank ($FM_{Sk}$) and joint moment with action on shank and reaction on thigh ($JM_{Sk-Th}$) for perturbation types P1-P5, as well as unperturbed gait (UP) as reference. Metrics reported are the step length of the perturbed limb ($SL_P$), and step length of the trailing limb ($SL_T$) of the recovery step. (TEX)

**S1 Fig. Kinematic response to perturbation 2 (P2).** Left column: Assistance via hip flexion of the perturbed limb. Conditions denoted by lines are unperturbed (UP), unassisted (UA), free moment with action on thigh ($FM_{Th}$), joint moment with action on thigh and reaction on trunk ($JM_{Th-Tk}$), and joint moment with action on thigh and reaction on contralateral thigh ($JM_{Th-Th}$). Right column: Assistance via knee extension of the perturbed limb. Conditions denoted by lines are unperturbed (UP), unassisted (UA), free moment with action on shank ($FM_{Sk}$), joint moment with action on shank and reaction on thigh ($JM_{Sk-Th}$). Individual plots show the trunk lean angle, hip flexion angle, and knee flexion angle, and applied moment $M$ as a function of time. Bar graphs at top of each column show the gait events of the unperturbed

gait cycle (GC), where TO-PL and HS-PL are toe-off and heel-strike events of the perturbed limb, and, TO-TL and HS-TL are toe-off and heel-strike events of the trailing limb.
(EPS)

**S2 Fig. Kinematic response to perturbation 3 (P3).** Left column: Assistance via hip flexion of the perturbed limb. Conditions denoted by lines are unperturbed (UP), unassisted (UA), free moment with action on thigh ($FM_{Th}$), joint moment with action on thigh and reaction on trunk ($JM_{Th-Tk}$), and joint moment with action on thigh and reaction on contralateral thigh ($JM_{Th-Th}$). Right column: Assistance via knee extension of the perturbed limb. Conditions denoted by lines are unperturbed (UP), unassisted (UA), free moment with action on shank ($FM_{Sk}$), joint moment with action on shank and reaction on thigh ($JM_{Sk-Th}$). Individual plots show the trunk lean angle, hip flexion angle, and knee flexion angle, and applied moment $M$ as a function of time. Bar graphs at top of each column show the gait events of the unperturbed gait cycle (GC), where TO-PL and HS-PL are toe-off and heel-strike events of the perturbed limb, and, TO-TL and HS-TL are toe-off and heel-strike events of the trailing limb.
(EPS)

**S3 Fig. Kinematic response to perturbation 5 (P5).** Left column: Assistance via hip flexion of the perturbed limb. Conditions denoted by lines are unperturbed (UP), unassisted (UA), free moment with action on thigh ($FM_{Th}$), joint moment with action on thigh and reaction on trunk ($JM_{Th-Tk}$), and joint moment with action on thigh and reaction on contralateral thigh ($JM_{Th-Th}$). Right column: Assistance via knee extension of the perturbed limb. Conditions denoted by lines are unperturbed (UP), unassisted (UA), free moment with action on shank ($FM_{Sk}$), joint moment with action on shank and reaction on thigh ($JM_{Sk-Th}$). Individual plots show the trunk lean angle, hip flexion angle, and knee flexion angle, and applied moment $M$ as a function of time. Bar graphs at top of each column show the gait events of the unperturbed gait cycle (GC), where TO-PL and HS-PL are toe-off and heel-strike events of the perturbed limb, and, TO-TL and HS-TL are toe-off and heel-strike events of the trailing limb.
(EPS)

**S4 Fig. Kinematic response of trailing limb to perturbation 4 (P4).** Left column: Assistance via hip flexion of the perturbed limb. Conditions denoted by lines are unperturbed (UP), unassisted (UA), free moment with action on thigh ($FM_{Th}$), joint moment with action on thigh and reaction on trunk ($JM_{Th-Tk}$), and joint moment with action on thigh and reaction on contralateral thigh ($JM_{Th-Th}$). Right column: Assistance via knee extension of the perturbed limb. Conditions denoted by lines are unperturbed (UP), unassisted (UA), free moment with action on shank ($FM_{Sk}$), joint moment with action on shank and reaction on thigh ($JM_{Sk-Th}$). Individual plots show the trunk lean angle, hip flexion angle, and knee flexion angle, and applied moment $M$ as a function of time. Bar graphs at top of each column show the gait events of the unperturbed gait cycle (GC), where TO-PL and HS-PL are toe-off and heel-strike events of the perturbed limb, and, TO-TL and HS-TL are toe-off and heel-strike events of the trailing limb.
(EPS)

**S1 Appendix. Additional considerations for practical application.**
(TEX)

## Acknowledgments

We would like to thank Thomas Geijtenbeek for assisting with the implementation of the tripping simulations in SCONE.

## Author Contributions

**Conceptualization:** Saher Jabeen, Patricia M. Baines, Jaap Harlaar, Heike Vallery.

**Data curation:** Saher Jabeen, Patricia M. Baines.

**Formal analysis:** Saher Jabeen, Patricia M. Baines, Andrew Berry.

**Funding acquisition:** Heike Vallery.

**Investigation:** Saher Jabeen, Patricia M. Baines.

**Methodology:** Saher Jabeen, Patricia M. Baines, Jaap Harlaar, Heike Vallery, Andrew Berry.

**Project administration:** Heike Vallery.

**Resources:** Heike Vallery.

**Software:** Saher Jabeen.

**Supervision:** Jaap Harlaar, Heike Vallery, Andrew Berry.

**Validation:** Saher Jabeen, Patricia M. Baines, Andrew Berry.

**Visualization:** Saher Jabeen, Patricia M. Baines, Andrew Berry.

**Writing – original draft:** Saher Jabeen, Patricia M. Baines, Andrew Berry.

**Writing – review & editing:** Saher Jabeen, Patricia M. Baines, Jaap Harlaar, Heike Vallery, Andrew Berry.

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
