## [Decision Letter · Decision Letter 0]

14 Jun 2022

PONE-D-22-01641Reaction moments matter when designing lower-extremity robots for tripping recoveryPLOS ONE

Dear Dr. Baines,

Thank you for submitting your manuscript to PLOS ONE. Firstly, we would like to apologize for the delay in processing your manuscript. It has been exceptionally difficult to secure reviewers to evaluate your study. We have now received two completed reviews, which are available below.

After careful consideration, we feel that it has merit but does not fully meet PLOS ONE’s publication criteria as it currently stands. Both Reviewers raised several scientific concerns about the current manuscript. Therefore, we invite you to submit a revised version of the manuscript that addresses the points raised during the review process. 

<ul1 style="margin-top:0;margin-bottom:0;padding-inline-start:48px;"> <li> 

 <li> 

 <li> 

We look forward to receiving your revised manuscript.

Kind regards,

Thomas Tischer

Staff Editor

PLOS ONE

Journal Requirements:

This research work was supported by the Netherlands Organisation for Scientific Research (NWO) Innovational Research Incentives Scheme Vidi grant 14865.

However, funding information should not appear in the Acknowledgments section or other areas of your manuscript. We will only publish funding information present in the Funding Statement section of the online submission form. 

This research work (HV,SJ) was supported by the Netherlands Organisation for Scientific Research (NWO) Innovational Research Incentives Scheme Vidi grant 14865. The funders had no role in study design, data collection and analysis, decision to publish, or preparation of the manuscript.

Reviewers' comments:

Reviewer's Responses to Questions

**Comments to the Author**

1. Is the manuscript technically sound, and do the data support the conclusions?

Reviewer #1: Yes

Reviewer #2: Yes

2. Has the statistical analysis been performed appropriately and rigorously? 

Reviewer #1: N/A

Reviewer #2: Yes

3. Have the authors made all data underlying the findings in their manuscript fully available?

Reviewer #1: Yes

Reviewer #2: Yes

4. Is the manuscript presented in an intelligible fashion and written in standard English?

Reviewer #1: Yes

Reviewer #2: Yes

5. Review Comments to the Author

Reviewer #1: This is the study that simulated the recovery process of the gait after tripping. The simulator applies recovery torque to the trunk, thigh, and shank. The effect of assisted links and assist torque on the performance of recovery motion, such as MoS and the number of recovery steps. Such parametric study is important to investigate the sensitiveness and mechanism of recovery motion. However, I have serious concern about the capability of proposed gait simulator for estimating the recovery motion. Furthermore, following points should be reconsidered.

Major

The application of free moment to the wearable device does not seems feasible. Although the reaction wheel, control moment gyro, and thruster are refereed, they were not used to assist specific body links such as thigh. Furthermore, as the principle of these mechanisms, sufficient mass is required to generate torque. It is a big disadvantage of the wearable device. In addition, the inertia of rotating body of these mechanisms will disturb the gait motion. I think the other actuator is required to apply free moment.

The evaluation index of recovery is not elaborate. The number of recovery steps counts the steps before fall. Thus, the number became 22 for all the trials that did not fall. It is difficult to evaluate the timing of recovery. It is required to determine the physical index of “balance recovery” and evaluate the difference of the timing of recovery among trials.

The validity of the gait algorithm is unclear. It is mentioned that the non-assisted gait cannot recover from the smallest perturbation. Perhaps, it is because the gait algorithm was not developed for simulating the recovery from tripping. However, of cause the healthy adult usually can recover from small perturbation without assistance. The analysis of recovery motion should consider the human’s general ability to recover from tripping.

The validity of gait simulator should be discussed. The qualitative and quantitative investigation should be done by comparing simulation output with the motion observed in the actual tripping experiment reported previously.

Minor

Generally, tripping experiments use vertical plate. Why was the spherical obstacle used?

The timing to apply assist torque seems unclear. How many gait cycles be the assist torque applied to?

The method to tune control parameters explained in Fig. 3 is ambiguous. What’s the objective function of this optimization?

Reviewer #2: This is an excellent paper and will make a significant contribution to the growing field of lower-extremity robotic aids. A few minor comments:

1. The context for the internal and external moments is unclear until pg 3 (Pg 9 of PDF) when the examples are given to the different types of gait assistive technologies (wearable devices verses stationary devices). It may aid the reader to have this context presented earlier in the Introduction.

2. Pg 8 of PDF, line 23: Perhaps, but falls in seniors are more precisely linked to transitions to (chair transfers) or from gait (turns, etc.). See work by Stephen Robinovitch at SFU.

3. Pg 17 of PDF, line 352: Interesting finding. Some recent studies of the Keeogo (BTemia) exoskeleton which is of the JM_Sk-Th type would appear to agree with this - both healthy individuals and patients do not walk faster when wearing the device.

4. Pg 18 of PDF, line 365: Wouldn't this be a knee exoskeleton?

6. PLOS authors have the option to publish the peer review history of their article (what does this mean?). If published, this will include your full peer review and any attached files.

Reviewer #1: No

Reviewer #2: No

---

## [Author Response · Author response to Decision Letter 0]

8 Oct 2022

For better readability please see letter in 'Response to Reviewers.pdf'. The plain text copy-pasted below:

Date October 8, 2022

Contact Patricia M. Baines

Phone +31 646117155

E-mail p.m.baines@tudelft.nl

Subject Revision article

Faculty of Mechanical, Maritime and

Materials Engineering,

Department of BioMechanical Engineering

Mekelweg 2 (Building 34)

2628 CD Delft

PO Box

Postbus 5

2600 AA Delft

PLOS ONE

Editorial Manager

Dear Dr. Tischer,

Thank you for your careful consideration of our manuscript. We found the suggestions of

the editor and reviewers to be helpful and highly insightful and would like to thank you

for the opportunity to submit a revised version of our manuscript.

Revisions in the manuscript are highlighted in blue text in ’Revised Manuscript with Track

Changes’, and below, you will find our responses to each comment in detail.

Journal Requirements:

Comment 1: Please ensure that your manuscript meets PLOS ONE’s style requirements,

including those for file naming.

Response: We have revisited the style requirements and ensured correct file naming.

Based on the provided links we made some minor changes on the title page and we have

used the LATEX PLOS ONE template found at

https://journals.plos.org/plosone/s/latex

Comment 2: Thank you for stating the following in the Acknowledgments Section of

your manuscript: “This research work was supported by the Netherlands Organisation for

Scientific Research (NWO) Innovational Research Incentives Scheme Vidi grant 14865.”

However, funding information should not appear in the Acknowledgments section or

other areas of your manuscript. We will only publish funding information present in the

Funding Statement section of the online submission form. Please remove any fundingrelated

text from the manuscript and let us know how you would like to update your

Funding Statement. Currently, your Funding Statement reads as follows: “This research

work (HV,SJ) was supported by the Netherlands Organisation for Scientific Research

(NWO) Innovational Research Incentives Scheme Vidi grant 14865. The funders had no

role in study design, data collection and analysis, decision to publish, or preparation of

the manuscript.” Please include your amended statements within your cover letter; we

will change the online submission form on your behalf.

Response: We agree with the information in the funding statement. Thank you for

providing it. We have removed funding information from the Acknowledgements section.

Comment 3: We note that you have stated that you will provide repository information

for your data at acceptance. Should your manuscript be accepted for publication, we will

hold it until you provide the relevant accession numbers or DOIs necessary to access your

data. If you wish to make changes to your Data Availability statement, please describe

these changes in your cover letter and we will update your Data Availability statement

to reflect the information you provide.

Response: Thank you for the information. A repository containing all the documented

simulation input files, the data extracted from simulation output, and the scripts to

extract and visualize the data can be accessed via a private link at 4TUD data repository

(doi:10.4121/21197149 will be activated once the repository is made public). These files

are available to use under the Apache 2.0 license.

Response to Reviewer 1:

Comment 1: This is the study that simulated the recovery process of the gait after tripping.

The simulator applies recovery torque to the trunk, thigh, and shank. The effect

of assisted links and assist torque on the performance of recovery motion, such as MoS

and the number of recovery steps. Such parametric study is important to investigate the

sensitiveness and mechanism of recovery motion.

Response: Thank you for the recognition of our efforts. This was indeed our intent.

Comment 2: The application of free moment to the wearable device does not seems

feasible. Although the reaction wheel, control moment gyro, and thruster are refereed,

they were not used to assist specific body links such as thigh. Furthermore, as the

principle of these mechanisms, sufficient mass is required to generate torque. It is a big

disadvantage of the wearable device. In addition, the inertia of rotating body of these

mechanisms will disturb the gait motion. I think the other actuator is required to apply

free moment.

Response: The reviewer’s concerns over the technical feasibility of external-reaction

wearable actuators are certainly valid. The section ‘Discussion: Practical application:

External-reaction wearable actuators’ has been rewritten to more explicitly address these

points:

... However, the principle of operation also introduces several limitations.

The presence of a passive mass for energy storage, and sometimes multiple

motors (control moment gyroscopes), can entail that inertial-manipulating

actuators are often larger and heavier than alternatives with similar power

output. While the detrimental effects of added mass on gait is similar for

all wearable devices, actuators with rotational energy storage (e.g., reaction

wheels and control moment gyroscopes) also have the potential to disturb gait

through unintended gyroscopic moments induced through rotational movement

of the actuated body segment – however, these induced moments can

be partly or wholly cancelled internally via mechanical constraints (Brown et

al. 2008) or control (Valk et al. 2018) or even be exploited to produce intentional

changes in gait patterns (Jabeen et al. 2021). Finally, although

all actuator types are limited in their maximum output moment and power,

inertia-manipulating actuators also have impulse constraints that limit the

duration of moment application and prevent them from providing continuous

support in static directions.

Given the relatively recent appearance of these technologies in wearable applications,

there are few empirical analyses and it is yet unclear which actuation

architecture is best (or even feasible) for practical implementation. For

example, most empirical studies involving control moment gyroscopes have

thus far sought to influence balance through control of the motion of the

CoM (Chiu et al. 2014, Lemus et al. 2020, Romtrairat et al. 2020), whereas

control of the lower extremities has been investigated only in simulations

(Jabeen et al. 2019). However, parallel research has explored the uses of

these actuators on the limbs for other purposes, such as tremor-suppression

(GyroGlove by GyroGear; London, UK), upper-extremity prostheses (Jarc et

al. 2006, Muller et al. 2017), and emulation of virtual environments (Duda et

al. 2015). Ongoing research continues to investigate the extent to which optimisation

of mechanical designs can mitigate technical limitations and exploit

the proposed advantages (Meijneke et al. 2021). While much development

and validation of these emerging technologies remains, they may soon fulfill

a complementary roll in the technological landscape and offer versatile

solutions for facilitating limb movement and balance.

At the Delft Biorobotics Laboratory at TU Delft, extensive research has been performed to

optimize the design of wearable gyroscopic actuators to address these technical challenges.

Comment 3: The evaluation index of recovery is not elaborate. The number of recovery

steps counts the steps before fall. Thus, the number became 22 for all the trials that did

not fall. It is difficult to evaluate the timing of recovery. It is required to determine the

physical index of “balance recovery” and evaluate the difference of the timing of recovery

among trials.

Response: Indeed, our selection of outcome measures needs further elaboration. The

binary presence/lack of a fall and the number of steps before a fall are certainly not comprehensive

measures of balance performance. To make our approach clear, we have added

substantial text to explain the reasoning for the primary outcomes and bring greater emphasis

to the role of the secondary outcome measures for interpreting responses in terms

of kinematic and kinetic movement characteristics.

In the case, gait is sustained (i.e., no fall occurs), there are several other measures of

disturbance-rejection available, such as the number of steps until periodicity, the Gait

Sensitivity Norm, and Lyapunov exponents of a Poincar´e section of the orbital dynamics,

among others. However, when the disturbance is not successfully rejected (i.e., the intervention

does not successfully prevent a fall), these measures have limited use. Instead,

it is relevant to consider the degree to which a fall is resisted. This is the case in the

present study since we wish to compare interventions for preventing falls.

To clarify which simulations successfully avoided falling, we have replaced the upper ceiling

of 22 steps with the designation ‘Full Recovery’ (FR). This is because the simulation

horizon is arbitrary, since any simulated perturbation followed by more than 5 steps would

always converge back to cyclical gait.

The section ’Methods: Performance measures’ has been modified to reflect these clarifications:

The primary criterion for assessing the efficacy of each intervention was

whether or not the intervention prevented a fall following a tripping event.

Although this is not a comprehensive metric of balance proficiency, it is a

practical and unambiguous measure of functional performance with a high

face validity for fall avoidance. Simulations in which a fall was successfully

avoided were designated ‘Full Recovery’ (FR).

When a fall is not successfully avoided, the characteristics of the response are

still relevant for analysis. Even if external assistance does not prevent a fall

in simulation, it can mitigate the effects of the perturbation and prolong gait

so that additional, real-world action could potentially be taken to prevent

injury or sustain gait – for example, affording the wearer more time to react

and execute their own postural response, complementary to the external assistance.

Therefore, the number of steps between the tripping event and the

falling event was recorded as a secondary outcome. Steps were counted with

respect to the first heel-strike after perturbation until either a body segment

other than the foot contacted the ground or the heel of the swing foot was

placed posteriorly to the toe of the stance foot.

and the number of steps for ‘Full Recovery’ is clarified in the section ‘Methods: Assistance’:

If the model continued to walk for at least ten steps after obstacle contact, it

was considered to have fully recovered – preliminary simulations showed that

falls always occurred within five steps of perturbation, similar to the number

of steps required by healthy young adults to normalize gait following tripping

(Forner Cordero et al. 2003).

We have also added further description of our secondary outcome measures, including

reference to prior studies that have made use of them. In addition to the existing text

explaining the role of the MoS to describe and interpret the physical response to tripping

and assistance, we have added the following text to describe our use of joint kinematics:

The kinematic responses in the first step following perturbation were also

analyzed. Kinematic features such as the trunk forward lean angle and joint

kinematics are widely used to differentiate between fallers and non-fallers

(van Die¨en et al. 2005, Pavol et al. 2001, Grabiner et al. 1993). Kinematic

responses are reported for a gait cycle starting at toe-off of the perturbed

(TO-PL) limb before perturbation and ending at toe-off of the same limb

after perturbation (Fig 1a). The trunk forward lean angle is measured from

the vertical axis and is reported as a positive angle when the body leans

forward. The hip flexion angle and knee flexion angle are presented along

with the lengths of the first steps of the perturbed (swing) limb (SLP) and

the trailing (non-perturbed) limb (SLT).

While a single, universal measure of balance performance unfortunately remains elusive,

we believe this combination of functional measures and kinematic/kinetic measures

sufficiently describes the principal features differentiating the different modes of lowerextremity

assistance.

Comment 4: The validity of the gait algorithm is unclear. It is mentioned that the

non-assisted gait cannot recover from the smallest perturbation. Perhaps, it is because

the gait algorithm was not developed for simulating the recovery from tripping. However,

of cause the healthy adult usually can recover from small perturbation without assistance.

The analysis of recovery motion should consider the human’s general ability to recover

from tripping.

Response: The gait algorithm has indeed been previously validated in a number of

empirical studies, including with tripping perturbations. The following text and references

have been added to the section ’Methods: Simulation tool’:

... The controller used to generate walking patterns for this study has been

widely validated in different studies to analyse aspects of impaired gait, such

as the effects of aging on balance control (Song et al. 2018, Reimann et al.

2020) and muscle activity (Ping et al. 2021). The model has also been validated

in response to mechanical disturbances, such as tripping by constraint

of the swing leg and slipping of the stance leg, yielding response trends similar

to experimental data for the majority of muscles (Song et al. 2017). ...

We clarify that the magnitudes of the perturbations were specifically set to induce falling

in both ‘Method: Simulation tool’:

Although the generated gait patterns were robustly stable to slight deviations,

the walking controller was not optimized to recover the balance from large

perturbations, and the model would fall if no intervention was made. This

represents a scenario in which a human would need support to avoid falling.

and ‘Methods: Perturbations’:

... To explore the efficacy of different interventions for preventing falls after

tripping, the perturbations were designed such that the model could not

sustain gait in the absence of external assistance.

However, a model is indeed an idealization of reality and may not perfectly the represent

the response to external moments applied to the lower extremities. This is stated in the

section ’Discussion: Study limitations and future work’:

The SCONE planar simulation model and underlying reflex model has been

previously validated against human data for unperturbed walking and certain

types of balance disturbances (Waterval et al. 2021). However, it is unclear

the extent to which dissimilarities in the experimental protocols (e.g., the

mode, magnitude, and timing of balance disturbance) and different choice of

outcome measures would affect the model validity. For example, trunk kinematic

responses are critical for distinguishing fallers from non-fallers during

tripping responses (Pavol et al. 2001), yet here the upper body was modelled

as a single segment, in which the pelvis, vertebrae, head, and arms were

rigidly fused together. Qualitative comparison of our simulation results (unassisted

condition) with empirical studies show a similar trend between peak

trunk lean angle and the magnitude of the tripping perturbation (Grabiner et

al. 1993, Graviner et al. 1996) and similar characteristics of the knee and

hip joint responses following tripping (Grabiner et al. 1993). Nevertheless,

a quantitative comparison is lacking and there is currently no data to validate

the neuromechanical response to external moments applied at different

locations on the body. It is thus far unclear whether moments produced by

wearable actuators could, under any circumstances, initiate additional, unmodelled

reflexes that might change the response characteristics. In addition,

a JM hip actuator would conventionally attach to the thigh and pelvis, but,

because the pelvis and trunk are fused in the simulation model, the simulation

may falsely exaggerate trunk lean when assistive moments are applied.

Comment 5: The validity of gait simulator should be discussed. The qualitative and

quantitative investigation should be done by comparing simulation output with the motion

observed in the actual tripping experiment reported previously.

Response: Both the reflex-based gait model and the SCONE simulation environment

have been validated in several studies. In addition to clarifying the validity and limitations

of the gait model (please see the response and quotations of the previous comment), we

have clarified the validity of the SCONE implementation in ‘Methods: Simulation tool’:

... The SCONE implementation of this model has also been validated against

normative data (Waterval et al. 2021). ...

A full list of publications using SCONE is available at

https://scone.software/doku.php?id=publications .

In the section ’Discussion: Study limitations and future work’ we also compare the simulated

kinematic responses to tripping with empirical studies – please see the last quote

in the response to the previous comment.

Comment 6: Generally, tripping experiments use vertical plate. Why was the spherical

obstacle used?

Response: This was one of the limitations of the SCONE simulation environment. At

the time of this study design, it was not possible to encode a step-like object to simulating

tripping, hence a spherical object was used instead. This limitation is now mentioned in

the section ’Methods: Perturbation’:

A spherical obstacle of varying diameter (shape constrained by the software

at the time of writing) was fixed to the ground surface to obstruct the swing

leg and produce tripping events (Fig 2).

Comment 7: The timing to apply assist torque seems unclear. How many gait cycles

be the assist torque applied to?

Response: The assistive torque was applied to only one gait cycle – the same cycle in

which the model is perturbed but after the foot has cleared the obstacle. The duration

of the assistance was fixed but the onset time was varied to maximize the number of

steps recovered following perturbation. This has been clarified in the section ’Methods:

Assistance’:

We tested 5 different types of assistance in which either a flexion moment on

the thigh or an extension moment on the shank was applied to assist forward

placement of the foot of the obstructed leg. This assistance was applied

unilaterally to the perturbed limb during the swing phase of the obstructed

step but after obstacle contact.

...

For comparing actuator types and placement, all moments were parametrized

as an ideal square wave of magnitude M, start time within the perturbed

swing phase ts, and sustained duration T (Fig. 1b). A range of values

of these three variables was simulated to determine the best parameters for

each use case. Considering the results obtained from a previous simulation

study (Jabeen et al. 2019) the duration of the assistance was set to 200ms,

peak moment limits [0,20] Nm with intervals of 2Nm, and start time is with

respect to foot and obstacle clearance. The assistance was initiated after the

foot cleared the obstacle, with start times in [0,210] ms (30ms intervals).

Comment 8: The method to tune control parameters explained in Fig. 3 is ambiguous.

What’s the objective function of this optimization?

Response: The objective function was not a continuous function, but rather a hierarchical

piece-wise set of rules. To select the control parameters, this rule set was evaluated

using an exhaustive grid search over a pre-defined discrete parameter space. We have

added clarification how we tuned the control parameters to the text of section ’Methods:

Assistance’:

To compare the best-case realization of each actuator type and placement,

an exhaustive grid search was performed to determine the control parameters

(assistance magnitude and start time) that maximized the number of steps

completed following the tripping event. The perturbation was applied halfway

through a walking simulation of approximately 30 s, in which the latter 15 s

for the model to get back in cyclic motion. If the model continued to walk

for at least ten steps after obstacle contact, it was considered to have fully

recovered – preliminary simulations showed that falls always occurred within

five steps of perturbation, similar to the number of steps required by healthy

young adults to normalize gait following tripping (Forner Cordero et al. 2003).

In the event of multiple candidates for a full recovery, the parameter set was

selected that first minimized the assistance magnitude and then minimized

assistance start time (Fig 3). In the case of partial or failed recovery (model

walked only a few steps and fell), the assistance parameter set was selected

with maximum anterior margin of stability (MoS) (Hof et al. 2005) of the

recovery step. Fig 2 shows the gait cycle and MoS definitions used for the

analysis and the gait events in terms of perturbed and recovery limb.

and modified Fig. 3:

Fig 3. Workflow for selection of best-case assistance parameters. Selection procedure

of assistance parameters magnitude M and start time ts based on the outcome

measures number of steps (n) and margin of stability (MoS) and the values of the assistance

parameters.

Response to Reviewer 2:

Comment 1: This is an excellent paper and will make a significant contribution to the

growing field of lower-extremity robotic aids.

Response: Thank you for your kind words.

Comment 2: The context for the internal and external moments is unclear until pg 3

(Pg 9 of PDF) when the examples are given to the different types of gait assistive

technologies (wearable devices verses stationary devices). It may aid the reader to have

this context presented earlier in the Introduction.

Response: We have reduced the amount of background information presented before

introducing robotic orthoses and different definitions of reaction moments. The first

mention of these now appear in the second paragraph.

Comment 3: Pg 8 of PDF, line 23: Perhaps, but falls in seniors are more precisely

linked to transitions to (chair transfers) or from gait (turns, etc.). See work by Stephen

Robinovitch at SFU.

Response: In community-dwelling adults, gait is cited as the activity with the largest

risk of falling (Blake et al. 1988, van Die¨en et al. 2005). However, elderly persons in

long-term care, such as in the studies of Robinovitch, are generally less mobile and tend

to be confined to indoor spaces. The text originally referred to has been removed, and

Paragraph 1 has been clarified to emphasise that we specifically focus on communitydwelling

persons with gait impairments. Paragraph 1 now reads as follows:

... Amongst community-dwelling individuals, more than 53% of falls occur

due to tripping while walking (Blake et al. 1988, van Die¨en et al. 2005) ...

Comment 4: Pg 17 of PDF, line 352: Interesting finding. Some recent studies of the

Keeogo (BTemia) exoskeleton which is of the JMSk-Th type would appear to agree with

this - both healthy individuals and patients do not walk faster when wearing the device.

Response: Thank you for referring us to this interesting finding. We have added the

following text to the paragraph in question:

< Reaction moments on the thigh initiate early foot contact and produce a

shorter step length >... This may also contribute to an empirical observation

that a powered knee orthosis with joint actuators significantly decreases gait

speed in both healthy and impaired users compared to when the device is not

worn (although it is unclear to what extent this may be due to the added

inertia or impedance of the device) (McGibbon et al. 2018).

Comment 5: Pg 18 of PDF, line 365: Wouldn’t this be a knee exoskeleton?

Response: Thank you for bringing this to our attention. This is indeed a typographical

error. The authors intended to refer to a hip exoskeleton, properly denoted as JMTh-Tk

instead of JMSk-Th. This is fixed in the manuscript:

... however when the reaction moment is applied on the trunk, as with a

traditional hip exoskeleton (JMTh-Tk), ...

If you have any questions or if anything is missing from our submission, please do not

hesitate to contact me.

With kind regards,

Patricia M. Baines

---

## [Decision Letter · Decision Letter 1]

8 Nov 2022

PONE-D-22-01641R1Reaction moments matter when designing lower-extremity robots for tripping recoveryPLOS ONE

Dear Dr. Baines,

Thank you for submitting your manuscript to PLOS ONE. After careful consideration, we feel that it has merit but does not fully meet PLOS ONE’s publication criteria as it currently stands. Therefore, we invite you to submit a revised version of the manuscript that addresses the points raised during the review process.Please ensure that your decision is justified on PLOS ONE’s publication criteria and not, for example, on novelty or perceived impact.

Please review the article given the reviewer's comments below to ensure that work in the literature is accurately reported. 

We look forward to receiving your revised manuscript.

Kind regards,

Aliah Faisal Shaheen

Academic Editor

PLOS ONE

Journal Requirements:

Additional Editor Comments:

Please make this final review of the article in relation to the comments made by the reviewer regarding the accuracy of representing previous studies.

Reviewers' comments:

Reviewer's Responses to Questions

**Comments to the Author**

1. If the authors have adequately addressed your comments raised in a previous round of review and you feel that this manuscript is now acceptable for publication, you may indicate that here to bypass the “Comments to the Author” section, enter your conflict of interest statement in the “Confidential to Editor” section, and submit your "Accept" recommendation.

Reviewer #1: All comments have been addressed

Reviewer #2: (No Response)

2. Is the manuscript technically sound, and do the data support the conclusions?

Reviewer #1: Yes

Reviewer #2: Yes

3. Has the statistical analysis been performed appropriately and rigorously? 

Reviewer #1: Yes

Reviewer #2: Yes

4. Have the authors made all data underlying the findings in their manuscript fully available?

Reviewer #1: Yes

Reviewer #2: Yes

5. Is the manuscript presented in an intelligible fashion and written in standard English?

Reviewer #1: Yes

Reviewer #2: Yes

6. Review Comments to the Author

Reviewer #1: (No Response)

Reviewer #2: Although the authors adequately addressed all the comments from this reviewer, it would behoove the authors to become more familiar with the paper they cited (McGibbon et al 2018) which is implied to include healthy subjects, but as far as I can tell it only includes people with multiple sclerosis. Also, the claim being made about "significantly reduced gait speed" is technically incorrect (the outcomes measures are functional tests: 6mwt, TUG test, and stair test). To be clear about my original comments: As far as this reviewer knows, no studies have shown an increase in walking speed whilst using this type of exoskeleton, but decrements in walking speed appear to vary depending on the population being studied.

7. PLOS authors have the option to publish the peer review history of their article (what does this mean?). If published, this will include your full peer review and any attached files.

Reviewer #1: No

Reviewer #2: No

---

## [Author Response · Author response to Decision Letter 1]

17 Dec 2022

Please see 'Response to reviewers.pdf' for full response.

---

## [Editor Report · Decision Letter 2]

21 Dec 2022

Reaction moments matter when designing lower-extremity robots for tripping recovery

PONE-D-22-01641R2

Dear Dr. Baines,

We’re pleased to inform you that your manuscript has been judged scientifically suitable for publication and will be formally accepted for publication once it meets all outstanding technical requirements.

Kind regards,

Aliah Faisal Shaheen

Academic Editor

PLOS ONE

---

## [Editor Report · Acceptance letter]

10 Feb 2023

PONE-D-22-01641R2 

Reaction moments matter when designing lower-extremity robots for tripping recovery 

Dear Dr. Baines:

I'm pleased to inform you that your manuscript has been deemed suitable for publication in PLOS ONE. Congratulations! Your manuscript is now with our production department. 

Kind regards, 

on behalf of

Dr. Aliah Faisal Shaheen 

Academic Editor

PLOS ONE